# Trends in alcohol consumption in middle-aged and older adults, assessed with self-report and the alcohol marker phosphatidylethanol – A longitudinal HUNT study

Kjerstin Tevik[1,2]*, Ragnhild Bergene Skråstad[3,4], Jūratė Šaltytė Benth[5,6], Geir Selbæk[1,7,8], Sverre Bergh[1,9], Olav Spigset[3,4], Steinar Krokstad[10,11], Anne-Sofie Helvik[1,2]

**1** Norwegian National Centre for Ageing and Health, Vestfold Hospital Trust, Tønsberg, Norway, **2** Department of Public Health and Nursing, Faculty of Medicine and Health Sciences, Norwegian University of Science and Technology (NTNU), Trondheim, Norway, **3** Department of Clinical Pharmacology, St. Olavs University Hospital, Trondheim, Norway, **4** Department of Clinical and Molecular Medicine, Faculty of Medicine and Health Sciences, Norwegian University of Science and Technology (NTNU), Trondheim, Norway, **5** Institute of Clinical Medicine, Campus Ahus, University of Oslo, Oslo, Norway, **6** Health Services Research Unit, Akershus University Hospital, Lørenskog, Norway, **7** Department of Geriatric Medicine, Oslo University Hospital, Oslo, Norway, **8** Institute of Clinical Medicine, Faculty of Medicine, University of Oslo, Oslo, Norway, **9** Research Centre for Age-Related Functional Decline and Disease, Innlandet Hospital Trust, Ottestad, Norway, **10** HUNT Research Centre, Department of Public Health and Nursing, Faculty of Medicine and Health Sciences, Norwegian University of Science and Technology (NTNU), Trondheim, Norway, **11** Levanger Hospital, Nord-Trøndelag Hospital Trust, Levanger, Norway

* kjerstin.e.tevik@ntnu.no

## Abstract

### Background

Alcohol is a leading risk factor for disease burden. We examined longitudinal trends in sex and age-specific alcohol consumption among middle-aged and older subjects who had participated in the population-based Trøndelag Health Study (HUNT) in Norway since the 1990s.

### Methods

This study included 23,151 individuals aged ≥43 years when they participated in the HUNT2 Survey (1995–1997) and who also had participated in the HUNT3 Survey (2006–2008), and/or the HUNT4 Survey (2017–2019). We used self-reported data to examine trends and identify sex- and age-specific differences in abstinence from alcohol, current drinking, risky drinking (≥8 units of alcohol/week), and heavy episodic drinking (≥5 or ≥6 units of alcohol in one sitting at least monthly). Concentrations of the objective alcohol marker phosphatidylethanol (PEth) were available in subsamples from HUNT3 to HUNT4.

**Data availability statement:** Due to restrictions imposed by the HUNT Research Centre (in accordance with the Norwegian Data Inspectorate), data cannot be made publicly available. Data are currently stored in the HUNT databank, and there are restrictions in place for the handling of HUNT data files. Data used from the HUNT Study in research projects will be made available on request to the HUNT Data Access Committee (kontakt@hunt.ntnu.no). The HUNT data access information (available here: http://www.ntnu.edu/hunt/data) describes in detail the policy regarding data availability.

**Funding:** This project was funded by the Swedish SAFF foundation (Stiftelsen Ansvar för fremtiden) (https://ansvarforframtiden.se/) through the Norwegian organization ACTIS (https://actis.no/). In addition, the project has been funded partly by the Norwegian National Centre for Ageing and Health (Ageing and Health), Vestfold Hospital Trust (https://www.aldringoghelse.no/english/). KT received the funding from the SAFF foundation and from Ageing and Health. This project has also been funded by the Clinic of Laboratory Medicine at the Department of Clinical Pharmacology at St. Olavs University Hospital in Trondheim (Norway) (PEth analyses) (https://www.stolav.no/) and the Norwegian DAM foundation (https://dam.no/) through the Norwegian non-profit organization 'Av og til' (https://avogtil.no/). RBS received the funding from St. Olavs University Hospital and the DAM foundation. The funders had no role in study design, data collection and analysis, decision to publish, or preparation of the manuscript.

**Competing interests:** The authors have declared that no competing interests exist.

## Results

The proportion of subjects with self-reported alcohol abstinence and with PEth concentrations <0.03 µmol/l increased from HUNT2 and/or HUNT3 to HUNT4, while heavy episodic drinking and PEth concentrations ≥0.03 µmol/l decreased from HUNT3 to HUNT4 in both sexes in most age groups, and more in men than in women. There was an increase in risky drinking from HUNT2 to HUNT4 in women and men aged 43–64 years in HUNT2. Men were more likely to consume alcohol than women measured with both self-report and with PEth in most age groups. Among those aged ≥65 years in HUNT2 a convergence between the sexes regarding abstinence and heavy episodic drinking was observed which was mostly caused by changes in men.

## Conclusion

Drinking patterns among middle-aged and older Norwegians have changed since the 1990s with a trend toward more abstinence and less heavy episodic drinking and PEth concentrations ≥0.03 µmol/l in both women and men with increasing age. There is a trend of more risky drinking with age among both sexes.

## Introduction

Throughout life, individuals change their alcohol consumption [1], and because alcohol consumption is a preventable cause of morbidity and mortality [2,3] it is important to follow trends in alcohol consumption. This knowledge can be used by health authorities and health care professionals to develop targeted intervention strategies with the aim of reducing alcohol consumption and alcohol-related harm in the population [1,4–6].

Given the progressive ageing of the Western population [7,8], we need updated knowledge about how different drinking patterns change from middle to older age. Knowledge about trends in alcohol consumption in later life is of particular importance because older adults are more susceptible to the negative health consequences from alcohol use compared to younger adults [9,10]. Older adults have, in general, reduced tolerance to alcohol because of lower first pass metabolism due to reduction in gastric alcohol dehydrogenase (ADH) and due to changes in body composition with increased body fat and decreased body water, leading to higher alcohol blood concentrations and longer effects of alcohol [9,10]. Moderate (1–2 units of alcohol/day) and higher levels of alcohol use (>2 units of alcohol/day) among middle-aged and older adults are related to several negative health consequences, including cancer [3,11], liver disease [3,12], cardiovascular disease [3,11,13], dementia [14,15], and death [3,11,12]. It is a well-documented connection between alcohol consumption and external harms such as risky driving and crash injuries [16] with the potential to cause severe harm, not only to themselves, but also to others, including pedestrians [16–18].

A common finding in previous research has been that middle-aged persons decrease their alcohol consumption as they age, however, research on this field has

shown conflicting results [19,20]. Poorer health, lower income, and loss of a life partner are significant factors associated with the decline in alcohol consumption [6,19,21]. However, several studies have shown that alcohol consumption (i.e., current drinking and frequent drinking) has been quite stable or has increased from middle to older age [1,4,20,22,23]. Good health, higher income, and more leisure time seem to be important factors for continuing regular alcohol consumption in later life [6,19]. Some individuals might also start drinking excessively and develop alcohol use disorders in older age [4,24]. A harmful drinking may be a way of coping with chronic disease, loss of a partner, and loss of work identity and meaning in life after retirement [25–27].

Previous cross-sectional studies have found that sex differences in alcohol consumption have diminished in recent decades [4,6,28,29], but middle-aged and older women are still more likely to abstain from alcohol, and they tend to drink less and less often compared to men [19,28,30,31]. Few longitudinal studies have examined the trends in sex differences in alcohol consumption among middle-aged and older women and men [4,22,23]. Two of these studies describe 10-year drinking trajectories among US women [22] and men [23] aged 50–65 years at baseline. The authors found that most of the participants had a stable drinking pattern as they aged, although, 4.9–8.8% of women [22] and 5.5% of men [23] increased their drinking. Another study from the US found that the prevalence of heavy episodic drinking increased among women ≥60 years over 17 years, while it was stable among men [4]. The increasing alcohol consumption in women is of concern because women are more sensitive to the negative effects of alcohol compared to men [32–34]. Women have lower first pass metabolism of alcohol due to reduced gastric ADH content and have also a lower volume of distribution due to increased body fat and less muscle mass, and thereby less body water than men. Thus, the risk of negative health consequences from equal amounts of alcohol is higher in women than men [32–35].

Self-reported alcohol consumption is susceptible to underestimation due to recall and reporting biases, especially among heavy drinkers [36,37]. The use of an objective alcohol marker can provide important additional information [38,39]. Of the objective markers of alcohol consumption, phosphatidylethanol 16:0/18:1 (PEth) measured in whole blood is one of the most specific, because PEth is only formed in the presence of ethanol [40,41]. Previous studies have found that PEth concentrations correlate well with the level of alcohol consumption and that PEth has a long detection time and is suitable to detect recent heavy alcohol consumption [38,39,42].

In Norway, findings from cross-sectional studies show that alcohol consumption among middle-aged and older women and men has increased over the last three decades [28–30,43]. However, we lack knowledge about trends in alcohol consumption in which the same individual is followed as they age from middle to older age. Use of data from the Trøndelag Health Study (HUNT), a large population-based longitudinal study conducted in Trøndelag County in Central Norway, gives us a unique opportunity to follow a whole population over several decades [44]. We also have data on PEth concentrations available from a subsample of the HUNT participants, which gives us the opportunity to evaluate alcohol consumption objectively. To our knowledge this is the first longitudinal study using PEth to examine changes in alcohol consumption in middle-aged and older women and men.

The first aim of the present study was to study trends in self-reported alcohol consumption over a period of 24 years among middle-aged and older women and men who had participated in at least two HUNT surveys (HUNT2 1995–1997 [≥43 years] and HUNT3 2006–2008 [≥54 years], and/or HUNT4 2017–2019 [≥65 years]) [44]. The second aim was to study changes in PEth concentrations over a period of 12 years among middle-aged and older women and men in HUNT3 (≥54 years) and HUNT4 (≥65 years). The third aim was to assess sex- and age-specific differences over time through self-reported alcohol consumption from HUNT2 to HUNT3 and HUNT4 and through PEth from HUNT3 to HUNT4.

## Materials and methods

### Study design

Data from three decennial population-based HUNT surveys (HUNT2, HUNT3, and HUNT4) were used to examine trends in alcohol consumption among middle-aged and older women and men [44].

## Study setting, data sources, and participants

The HUNT Study is a large population-based cohort study conducted in Trøndelag County in Central Norway, which consists of two regions: Nord-Trøndelag and Sør-Trøndelag [44]. This study used data from the Nord-Trøndelag region, which is considered to be fairly representative of the whole of Norway regarding geography, industry, age, sex, health status, and mortality, and follow international Western trends in public health regarding heart disease, blood pressure, body mass index, and smoking [45–47]. However, compared to Norway as a whole, Nord-Trøndelag lacks large cities, has lower numbers of immigrants, and the inhabitants have lower income and lower educational level [45,48,49].

So far, four HUNT surveys have been conducted: HUNT1 (1984–1986), HUNT2 (1995–1997), HUNT3 (2006–2008), and HUNT4 (2017–2019) [49,50]. Every resident aged 20 years or more has been invited to participate since HUNT1. In the present study, we used data from HUNT2, HUNT3, and HUNT4. The participation rate was 69.5% in HUNT2 (65,237 of 93,898 invited), 54.1% in HUNT3 (50,807 of 93,860 invited), and 54.0% in HUNT4 (56,042 of 103,800 invited) [45,49,50]. Individuals who participated in HUNT2, were recruited from August 15, 1995, to June 18, 1997; in HUNT3, from October 3, 2006, to June 25, 2008; and in HUNT4, from May 5, 2017, to February 21, 2019.

We included non-institutionalized adults who were 43 years or older when they participated in HUNT2 and later participated in HUNT3 (≥54 years) and/or HUNT4 (≥65 years). The participation rates among those aged ≥40 years in the last three HUNT surveys are described in detail in previous studies [45,49,50]. However, the participation rate was higher among women than men, was highest among individuals aged 60–69 years (85.6% in HUNT2, 71.1% in HUNT3, and 66.8% in HUNT4), and decreased with increasing age in both HUNT2, HUNT3, and HUNT4 [45,49,50].

In each HUNT survey the participants responded to two self-report questionnaires (Q1 and Q2). Interviews, clinical examinations, and laboratory measurements were performed at an examination station. Full details of the HUNT Study are presented elsewhere [45,49,50], and the questionnaires (Q1 and Q2) can be downloaded from the following webpage: https://www.ntnu.edu/hunt/data/que.

Data on income at an individual level were retrieved from Statistics Norway (SSB) [51].

## Measures

**Alcohol consumption.** The HUNT2, HUNT3, and HUNT4 surveys included self-reported questions about episodic drinking. Table 1 defines the drinking categories used to examine trends in self-reported alcohol consumption in our study, namely abstinence, current drinking, risky drinking, and heavy episodic drinking. The question about abstinence in HUNT2 ("Do you entirely abstain from alcohol?") differed from the question about abstinence in HUNT3 and HUNT4 ("I have never consumed alcohol" and "Not consumed alcohol at all last year") [52]. The questions about the volume of alcohol consumption (number of glasses of wine, beers, and spirits usually consumed during the last 14 days), have been the same in the last three HUNT surveys. Heavy episodic drinking is only assessed in HUNT3 and HUNT4. The participants in HUNT3 responded to the question on how often they drank 5 glasses of wine, beer, or spirits in one sitting, while in HUNT4, this question was changed to how often they drank 6 glasses of wine, beer, or spirits in one sitting. S1 Table describes in detail how the questions and definitions regarding abstinence, current drinking, and heavy episodic drinking have changed from HUNT2 to HUNT3 and HUNT4.

In a subsample in HUNT3 and HUNT4, PEth was analyzed in stored blood from the HUNT-biobank [39,53] (see Table 1). The subsample was randomly selected among participants who had answered at least one question regarding their alcohol consumption in any of the questionnaires, but independent of their reported alcohol consumption.

Although there is no unequivocal international consensus on how to interpret PEth concentrations, an international group of researchers with extensive experience in the field has published a consensus report suggesting that a PEth concentration of <20 ng/ml (0.028 μmol/l) is compatible with abstinence or low alcohol consumption [54]. A PEth concentration of 200 ng/ml (0.28 μmol/l) or greater is suggested to be strongly associated with chronic excessive alcohol consumption corresponding to an average consumption of 60 g or more of pure ethanol on a single drinking day over a prolonged

**Table 1. Definition of self-reported drinking categories and classification of alcohol consumption assessed by blood phosphatidylethanol (PEth) concentrations.**

| Self-reported drinking categories | Definition | Description[1] |
|---|---|---|
| Abstinence | HUNT2, HUNT3, and HUNT4 Total abstinence (HUNT2)/ Never consumed alcohol in lifetime or not consumed alcohol in the last year (HUNT3, HUNT4) | In HUNT2, participants who reported "yes" to the question "Do you entirely abstain from alcohol" were defined as abstainers. In HUNT3 and HUNT4, those who reported "never consumed alcohol" or "not consumed alcohol at all in the last year" in the drinking frequency questionnaire, were defined as abstainers. |
| Current drinking | HUNT2, HUNT3, and HUNT4 Consumed alcohol ≥ once per month (HUNT2) or ≥ few times per year (HUNT3, HUNT4) | In HUNT2, participants who reported drinking once a month or more in the drinking frequency questionnaire were defined as current drinkers. In HUNT3, those who reported "drinking a few times per year" or reported a higher drinking frequency category were defined as current drinkers. In HUNT4, those who reported drinking "once per month or less" or responded to a higher drinking frequency category were defined as current drinkers. |
| Risky drinking | HUNT2, HUNT3, and HUNT4 Drinking ≥8 units of alcohol per week | The participants in HUNT2, HUNT3, and HUNT4 reported the number of glasses of beer, wine, or spirits they usually consumed during a course of two weeks. In the present study, one glass of beer, wine, or spirits was defined as equivalent to one unit of alcohol. One unit of alcohol in Norway is defined as about 12 g of pure alcohol, which corresponds to one small beer (0.33 L) with 4.5% alcohol by volume (ABV), one glass of wine (0.125 L) with 12% ABV, and one drink of spirits (0.04 L) with 40% ABV [57,58]. The total number of units of beer, wine, or spirits usually consumed in two weeks were converted to the total number of units of alcohol consumed in one week. Risky drinking was defined as drinking ≥8 units of alcohol in a week. This definition is in line with alcohol guidelines for older adults in the US [59] and has been used in several epidemiological studies assessing risky drinking in older adults [60]. |
| Heavy episodic drinking | HUNT3 and HUNT4 Drinking ≥5 (HUNT3) or ≥6 (HUNT4) units of alcohol in one sitting at least monthly | In HUNT3 and HUNT4, participants who reported drinking ≥5 (HUNT3) and ≥6 (HUNT4) glasses of beer, wine, or spirits in one sitting at least monthly were defined as heavy episodic drinkers [60]. Glasses of beer, wine, or spirits were converted to units of alcohol. |
| **Alcohol consumption measured by PEth** | **Classification** | |
| | HUNT3 and HUNT4 Categorized PEth concentrations [38]: 1) <0.03 µmol/l 2) 0.03–0.09 µmol/l 3) 0.10–0.29 µmol/l 4) ≥0.30 µmol/l | PEth was analyzed in a subsample in HUNT3 (≥54 years: n = 12,847) and HUNT4 (≥65 years: n = 7,290). PEth is a specific alcohol marker that is only formed in the presence of ethanol and reflects the alcohol intake during the last 2–4 weeks prior to sampling [56,61]. Blood samples were drawn into 3 ml EDTA tubes and placed in refrigerated storage overnight. They were then frozen and stored at –80 °C until analysis [62]. A validated ultra-performance liquid chromatography tandem mass spectrometry (UPLC®-MSMS) method [63] was used to analyze PEth in whole blood. The quantification range was 0.03–4.0 µmol/l [63]. To convert to ng/ml use a factor of 703. |

Abbreviations: HUNT = Trøndelag Health Study; PEth = phosphatidylethanol 16:0/18:1.

[1]The HUNT questionnaires used to define and describe different drinking categories can be downloaded from https://www.ntnu.edu/hunt/data/que, and the changes in the questions regarding abstinence, current drinking, and heavy episodic drinking from HUNT2 to HUNT3 and HUNT4 are described in S1 Table.

duration for men and 40 g for women [54]. The consensus statement is in many respects in line with previous proposals for the interpretation of PEth concentrations [55,56], and the cut-offs are fairly well established. Because the limit of 0.30 µmol/l corresponds to a rather high level of alcohol consumption, it has been proposed to introduce 0.10 µmol/l as an additional cut-off to also identify those who have a somewhat lower, but still harmfully high level of alcohol consumption. This cut-off has been suggested to identify those who have an average daily alcohol consumption of 2–3 units of alcohol and has shown an acceptable sensitivity and specificity [38]. It is particularly relevant in the context of the drinking habits of older adults because they are more vulnerable to the harmful effects of alcohol [10]. In the present study, changes in different concentration intervals of PEth were examined (see Table 1).

**Socioeconomic and demographic variables.** The socioeconomic and demographic variables included were sex, age at the time of survey participation, level of education (up to 10 years of education, vocational and general education, or college and university), marital status (living with a spouse or a partner versus not), smoking (never smoked, previously smoked daily, or daily smoker), living place (urban or rural living), and income after taxes (Norwegian kroner). For the analyses, age was categorized into three groups in HUNT2 (43–53, 54–64, and ≥65 years=youngest), HUNT3 (54–64, 65–75, and ≥76 years=middle-aged), and HUNT4 (65–75, 76–86, and ≥87 years=oldest). Except for the measures of PEth and information about income from the SSB [51], all data from the HUNT surveys were based on self-reports from the participants.

## Ethics

The HUNT Study participants signed an informed and written consent allowing the use of their data for future medical research and allowing the linking of their data to other health and administrative registries, such as the SSB [45,49,50]. All personal identification data (names and personal ID numbers) were removed from the data files to ensure anonymity.

The HUNT research project is carried out according to the principles expressed in the Declaration of Helsinki and in accordance with the Norwegian Data Inspectorate and the Regional Committee of Medical and Health Research Ethics (REC) [44]. REC (reference number 407997), and the Norwegian Social Science Data Services (reference number 419689) approved the study. The analysis of PEth in the HUNT Study has been approved by the REC (REC ID 2017/1499) and has been thoroughly discussed in the HUNT ethical committee.

## Statistical analyses

Sample characteristics were described as means and standard deviations (SDs) and/or medians and quartiles for continuous variables and as frequencies and percentages for categorical variables. The prevalence of self-reported drinking categories and categorized PEth concentrations was presented for the entire sample as well as stratified by sex and age groups. Sex- and age-specific differences in change in dichotomous outcome variables were assessed by generalized linear mixed models with logit link function and random effects for participants to properly adjust for within-participant correlations. The models included fixed effects for time dummies (HUNT2, HUNT3, and HUNT4), sex (men vs. women) and categorized age (43–53/ 54–64/ 65–75, 54–64/ 65–75/ 76–86, and ≥65/ ≥76/ ≥87), as well as all two- and three-way interactions. A significant three-way interaction would imply that there are significant sex- and age-specific differences in alcohol consumption over time. To make the interpretation easier, the odds at each time point within each group as well as the odds ratios for group comparison were extracted from the model in a post-hoc analysis, and tabulated and illustrated graphically together with the corresponding 95% confidence intervals (CIs).

All analyses were performed in STATA v18. The results with p-values below 0.05 were considered statistically significant. All tests were two-sided.

## Results

Fig 1 shows in a flow diagram the number of participants included from HUNT2, HUNT3, and HUNT4 and number of participants with measured PEth in HUNT3 and HUNT4.

Table 2 shows the basic characteristics of the participants in HUNT2 (≥43 years; n=23,151; 53.3% women); HUNT3 (≥54 years; n=21,736, 53.7% women); and HUNT4 (≥65 years; n=14,667; 53.8% women). The mean age (SD) was 55.5 (8.8) years in HUNT2, 66.9 (8.8) years in HUNT3, and 74.5 (6.8) years in HUNT4.

### Prevalence and trends in self-reported alcohol consumption

The prevalence of abstinence in women and men in HUNT2 (≥43 years) was 15.4% and 7.2%, it increased to 16.8% and 8.7% in HUNT3 (≥54 years), and to 25.7% and 13.2% in HUNT4 (≥65 years), respectively (Table 2). This increasing trend

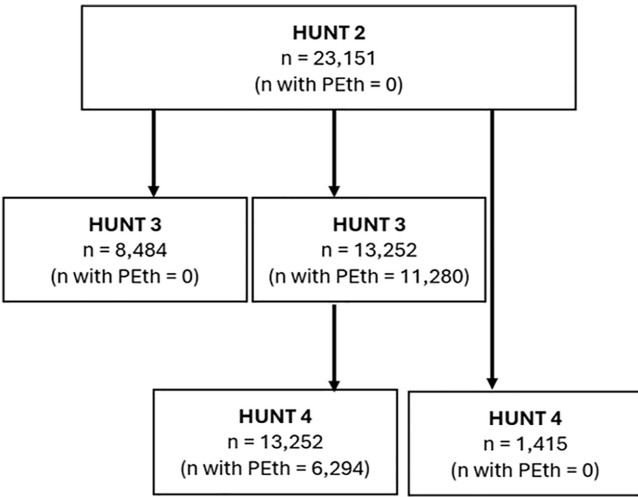

**Fig 1. Flow diagram.** Number of participants included from HUNT2, HUNT3, and/or HUNT4 and number of participants with measured PEth in HUNT3 and HUNT4.

in abstinence was significant in both sexes in the oldest age group from HUNT2 to HUNT3, and in all age groups from HUNT2 to HUNT4 and from HUNT3 to HUNT4 (Table 3 and Fig 2).

The prevalence of current drinking among women and men was 59.8% and 79.5% in HUNT2 (≥43 years), it increased to 83.2% and 91.3% in HUNT3 (≥54 years), and it decreased to 74.3% and 86.8% in HUNT4 (≥65 years), respectively (Table 2). The increasing trend in current drinking from HUNT2 to HUNT3 and the decreasing trend from HUNT3 to HUNT4 were significant among both women and men in all age groups (Table 3 and Fig 2).

Even though the prevalence of risky drinking was low, it increased slightly among both women and men from 0.8% and 4.4% in HUNT2 (≥43 years), to 1.8% and 6.6% in HUNT3 (≥54 years), and to 2.2% and 8.1% in HUNT4 (≥65 years), respectively (Table 2). This increasing trend in risky drinking was significant among both women and men from HUNT2 to HUNT3 and from HUNT2 to HUNT4 in the two youngest age groups, but only among men in the youngest age group from HUNT3 to HUNT4 (Table 3 and Fig 2). It was not sufficient data available in the oldest age group of women in HUNT4 to perform the analyses for risky drinking.

The prevalence of heavy episodic drinking was 6.0% among women in HUNT3 (≥54 years) but was considerably higher (23.5%) among men in HUNT3 (≥54 years), and it was lower in both sexes in HUNT4 (≥65 years) (2.3% in women and to 6.8% in men) (Table 2). This decrease from HUNT3 to HUNT4 in heavy episodic drinking was significant in all age groups for men, but only for the two youngest age groups for women (Table 3 and Fig 2).

### Sex- and age-specific differences in self-reported alcohol consumption

Women were more likely to abstain from alcohol than men in all age groups in HUNT2, HUNT3, and HUNT4, while men were more likely to be current drinkers, risky drinkers, and heavy episodic drinkers than women in HUNT2, HUNT3, and HUNT4 in all age groups (Table 3). The exception was for heavy episodic drinking in the oldest age group in HUNT4, where we found no significant difference between women and men.

The increasing trend in abstinence from HUNT2 to HUNT4 and the decrease in heavy episodic drinking from HUNT3 to HUNT4 were significantly more pronounced among men than women in the oldest age group when compared to the younger age group (Table 3). We found no other sex- or age-specific differences.

**Table 2. Overall sample characteristics of women and men ≥ 43 years at HUNT2 (1995–1997), ≥ 54 years at HUNT3 (2006–2008), and ≥65 years at HUNT4 (2017–2019), stratified by age. Numbers are frequencies and percentages (%) unless stated otherwise. Subjects participating in HUNT2 and HUNT3 and/or HUNT4 included.**

| Characteristics | HUNT2 1995–1997 (≥43 years) | | | HUNT3 2006–2008 (≥54 years) | | | HUNT4 2017–2019 (≥65 years) | | |
|---|---|---|---|---|---|---|---|---|---|
| | Women (n=12,351) | Men (n=10,800) | Total (n=23,151) | Women (n=11,669) | Men (n=10,067) | Total (n=21,736) | Women (n=7,898) | Men (n=6,769) | Total (n=14,667) |
| *Age (years)* Mean (SD) | 55.7 (9.0) | 55.2 (8.6) | 55.5 (8.8) | 67.1 (9.0) | 66.7 (8.6) | 66.9 (8.8) | 74.8 (7.0) | 74.1 (6.6) | 74.5 (6.8) |
| *Age groups* HUNT2/ HUNT3/ HUNT4 43–53  54–64  65–75 54–64  65–75  76–86 ≥65      ≥76      ≥87 | 6,247 (50.6) 3,852 (31.2) 2,252 (18.2) | 5,659 (52.4) 3,420 (31.7) 1,721 (15.9) | 11,906 (51.4) 7,272 (31.4) 3,973 (17.2) | 5,691 (48.8) 3,751 (32.1) 2,227 (19.1) | 5,041 (50.1) 3,303 (32.8) 1,723 (17.1) | 10,732 (49.4) 7,054 (32.5) 3,950 (18.2) | 5,014 (63.5) 2,342 (29.7) 542 (6.9) | 4,486 (66.3) 1,944 (28.7) 339 (5.0) | 9,500 (64.8) 4,286 (29.2) 881 (6.0) |
| *Education* [a] n Up to ten years Vocational and general College/university | 11,860 6,022 (50.8) 3,800 (32.0) 2,038 (17.2) | 10,523 3,739 (35.5) 4,420 (42.0) 2,364 (22.5) | 22,383 9,761 (43.6) 8,220 (36,7) 4,402 (19.7) | No data | No data | No data | 7,786 2,558 (32.9) 3,494 (44.9) 1,734 (22.3) | 6,715 1,351 (20.1) 3,397 (50.6) 1,967 (29.3) | 14,501 3,909 (27.0) 6,891 (47.5) 3,701 (25.5) |
| *Marital status* [a] n No living spouse or partner Living spouse or partner | 12,327 2,860 (23.2) 9,467 (76.8) | 10,790 1,926 (17.8) 8,864 (82.2) | 23,117 4,786 (20.7) 18,331 (79.3) | 11,662 4,382 (37.6) 7,280 (62.4) | 10,064 2,310 (23.0) 7,754 (77.0) | 21,726 6,692 (30.8) 15,034 (69.2) | 7,894 3,357 (42.5) 4,537 (57.5) | 6,763 1,639 (24.2) 5,124 (75.8) | 14,657 4,996 (34.1) 9,661 (65.9) |
| *Living in* [a] n Urban areas Rural areas | 12,351 7,773 (62.9) 4,578 (37.1) | 10,800 6,719 (62.2) 4,081 (37.8) | 23,151 14,492 (62.6) 8,659 (37.7) | 11,559 7,282 (63.0) 4,277 (37.0) | 9,991 6,209 (62.1) 3,782 (37.9) | 21,550 13,491 (62.6) 8,059 (37.4) | 7,897 5,074 (64.3) 2,823 (35.7) | 6,768 4,326 (63.9) 2,442 (36.1) | 14,665 9,400 (64.1) 5,265 (35.9) |
| *Smoking* [a] n Never smoked Previously smoked daily Daily smoker | 12,200 5,738 (47.0) 3,204 (26.3) 3,258 (26.7) | 10,725 3,520 (32.8) 4,426 (41.3) 2,779 (25.9) | 22,925 9,258 (40.4) 7,630 (33.3) 6,037 (26.3) | 11,121 4,949 (44.5) 3,795 (34.1) 2,377 (21.4) | 9,761 3,052 (31.3) 4,763 (48.8) 1,946 (19.9) | 20,882 8,001 (38.3) 8,558 (41.0) 4,323 (20.7) | 7,796 3,157 (40.5) 3,950 (50.7) 689 (8.8) | 6,730 2,259 (33.6) 4,040 (60.0) 431 (6.4) | 14,526 5,416 (37.3) 7,990 (55.0) 1,120 (7.7) |
| *Income (NOK)* [a,b] n Mean (SD) Median (Q1; Q3) | 12,201 112,033 (53,905) 109,139 (74,806; 145,854) | 10,798 173,851 (94,370) 165,273 (74,810; 145,854) | 22,999 141,057 (81,670) 136,822 (94,468; 174,984) | 11,653 185,321 (84,150) 172,162 (129,82; 226,357) | 10,065 266,546 (196,22) 240,233 (188,48; 303,949) | 21,718 222,964 (152,58) 202,428 (152,36; 266,258) | 7,895 263,648 (96,605) 245,854 (207,68; 298,16) | 6,765 352,168 (309,97) 306,773 (261,24; 371,58) | 14,660 304,497 (226,49) 273,281 (227,55; 333,043) |
| **Self-reported drinking categories** | | | | | | | | | |
| *Abstinence* [a,c] Age groups HUNT2/ HUNT3/ HUNT4 n 43–53  54–64  65–75 54–64  65–75  76–86 ≥65      ≥76      ≥87 All    All    All | 12,128 566 (9.1) 648 (17.2) 650 (30.0) 1,864 (15.4) | 10,693 299 (5.3) 263 (7.8) 206 (12.2) 768 (7.2) | 22,821 865 (7.3) 911 (12.7) 856 (22.2) 2,632 (11.5) | 11,024 550 (9.9) 651 (18.4) 650 (34.0) 1,851 (16.8) | 9,828 274 (5.5) 293 (9.0) 288 (17.7) 855 (8.7) | 20,852 824 (7.8) 944 (13.9) 938 (26.5) 2,706 (13.0) | 7,630 877 (17.9) 793 (35.7) 288 (58.2) 1,958 (25.7) | 6,623 418 (9.5) 333 (17.7) 122 (37.8) 873 (13.2) | 14,253 1,295 (13.9) 1,126 (27.4) 410 (50.1) 2,831 (19.9) |
| *Current drinking* [a,d] Age groups HUNT2/ HUNT3/ HUNT4 n 43–53  54–64  65–75 54–64  65–75  76–86 ≥65      ≥76      ≥87 All    All    All | 9,246 3,558 (68.7) 1,502 (54.6) 472 (36.0) 5,532 (59.8) | 9,257 4,289 (84.6) 2,232 (77.5) 838 (64.1) 7,359 (79.5) | 18,503 7,847 (76.5) 3,734 (66.3) 1,310 (50.0) 12,891 (69.7) | 11,024 5,016 (90.1) 2,893 (81.6) 1,264 (66.0) 9,173 (83.2) | 9,828 4,685 (94.5) 2,948 (91.0) 1,340 (82.3) 8,973 (91.3) | 20,852 9,701 (92.2) 5,841 (86.1) 2,604 (73.5) 18,146 (87.0) | 7,630 4,034 (82.1) 1,431 (64.3) 207 (41.8) 5,672 (74.3) | 6,623 3,996 (90.5) 1,553 (82.3) 201 (62.2) 5,750 (86.8) | 14,253 8,030 (86.1) 2,984 (72.6) 408 (49.9) 11,422 (80.1) |
| *Risky drinking* [a,e] Age groups HUNT2/ HUNT3/ HUNT4 n 43–53  54–64  65–75 54–64  65–75  76–86 ≥65      ≥76      ≥87 All    All    All | 11,532 58 (1.0) 23 (0.6) 7 (0.3) 83 (0.8) | 10,195 273 (5.1) 138 (4.3) 42 (2.6) 453 (4.4) | 21,727 331 (2.9) 161 (2.4) 49 (1.3) 541 (2.5) | 10,161 116 (2.2) 56 (1.7) 9 (0.5) 181 (1.8) | 9,241 390 (8.1) 172 (5.7) 48 (3.4) 610 (6.6) | 19,402 506 (5.0) 228 (3.7) 57 (1.8) 791 (4.1) | 6,294 109 (2.6) 27 (1.6) 0 136 (2.2) | 6,024 386 (9.3) 94 (5.8) 7 (2.9) 487 (8.1) | 12,318 495 (5.9) 121 (3.6) 7 (1.2) 623 (5.1) |

*(Continued)*

**Table 2.** (Continued)

| Characteristics | HUNT2 1995–1997 (≥43 years) | | | HUNT3 2006–2008 (≥54 years) | | | HUNT4 2017–2019 (≥65 years) | | |
|---|---|---|---|---|---|---|---|---|---|
| | Women (n = 12,351) | Men (n = 10,800) | Total (n = 23,151) | Women (n = 11,669) | Men (n = 10,067) | Total (n = 21,736) | Women (n = 7,898) | Men (n = 6,769) | Total (n = 14,667) |
| *Heavy episodic drinking* [a,f] *Age groups* HUNT2/ HUNT3/ HUNT4 n 43–53  54–64  65–75 54–64  65–75  76–86 ≥65  ≥ 76  ≥ 87 All  All  All | No data | No data | No data | 10,757 471 (8.7) 137 (4.0) 34 (1.8) 642 (6.0) | 9,547 1,594 (32.8) 525 (16.6) 123 (8.0) 2,242 (23.5) | 20,304 2,065 (20.1) 662 (10.0) 157 (4.6) 2,884 (14.2) | 7,224 122 (2.6) 36 (1.8) 5 (1.2) 163 (2.3) | 6,504 375 (8.6) 63 (3.4) 2 (0.7) 440 (6.8) | 13,728 497 (5.5) 99 (2.6) 7 (1.0) 603 (4.4) |
| **Categorized PEth concentrations** | | | | | | | | | |
| *PEth < 0.03 μmol/l* [a] *Age groups* HUNT2/ HUNT3/ HUNT4 n 43–53  54–64  65–75 54–64  65–75  76–86 ≥65  ≥ 76  ≥ 87 All  All  All | No data | No data | No data | 5,926 1,853 (62.8) 1,445 (75.9) 963 (89.9) 4,261 (71.9) | 5,354 1,218 (45.4) 1,018 (57.3) 676 (75.5) 2,912 (54.4) | 11,280 3,071 (54.5) 2,463 (66.9) 1,639 (83.4) 7,173 (63.6) | 3,383 1,464 (67.9) 870 (87.1) 219 (96.1) 2,553 (75.5) | 2,911 1,041 (54.0) 640 (76.6) 132 (88.6) 1,813 (62.3) | 6,294 2,505 (61.4) 1,510 (82.3) 351 (93.1) 4,366 (69.4) |
| *PEth 0.03–0.09 μmol/l* [a] *Age groups* HUNT2/ HUNT3/ HUNT4 n 43–53  54–64  65–75 54–64  65–75  76–86 ≥65  ≥ 76  ≥ 87 All  All  All | No data | No data | No data | 5,926 595 (20.2) 262 (13.8) 73 (6.8) 930 (15.7) | 5,354 726 (27.1) 373 (21.0) 117 (13.1) 1,216 (22.7) | 11,280 1,321 (23.5) 635 (17.2) 190 (9.7) 2,146 (19.0) | 3,383 412 (19.1) 81 (8.1) 8 (3.5) 501 (14.8) | 2,911 446 (23.2) 111 (13.3) 10 (6.7) 567 (19.5) | 6,294 858 (21.0) 192 (10.5) 18 (4.8) 1,068 (17.0) |
| *PEth 0.10–0.29 μmol/l* [a] *Age groups* HUNT2/ HUNT3/ HUNT4 n 43–53  54–64  65–75 54–64  65–75  76––86 ≥65  ≥ 76  ≥ 87 All  All  All | No data | No data | No data | 5,926 341 (11.6) 117 (6.1) 25 (2.3) 483 (8.2) | 5,354 505 (18.8) 235 (13.2) 71 (7.9) 811 (15.1) | 11,280 846 (15.0) 352 (9.6) 96 (4.9) 1,294 (11.5) | 3,383 205 (9.5) 35 (3.5) 1 (0.4) 241 (7.1) | 2,911 306 (15.9) 71 (8.5) 5 (3.4) 382 (13.1) | 6,294 511 (12.5) 106 (5.8) 6 (1.6) 623 (9.9) |
| *PEth ≥ 0.30 μmol/l* [a] *Age groups* HUNT2/ HUNT3/ HUNT4 n 43–53  54–64  65–75 54–64  65–75  76–86 ≥65  ≥ 76  ≥ 87 All  All  All | No data | No data | No data | 5,926 161 (5.5) 81 (4.3) 10 (0.9) 252 (4.3) | 5,354 232 (8.7) 152 (8.5) 31 (3.5) 415 (7.8) | 11,280 393 (7.0) 233 (6.3) 41 (2.1) 667 (5.9) | 3,383 75 (3.5) 13 (1.3) 0 88 (2.6) | 2,911 133 (6.9) 14 (1.7) 2 (1.3) 149 (5.1) | 6,294 208 (5.1) 27 (1.5) 2 (0.5) 237 (3.8) |

Abbreviations: SD = Standard deviation; HUNT = Trøndelag Health study; NOK = Norwegian kroner; n = number; PEth = phosphatidylethanol 16:0/18:1; Q1 = first quartile; Q3 = third quartile.

All in HUNT2 = All ≥ 43 years in HUNT2; All in HUNT3 = All ≥ 54 years in HUNT3; All in HUNT4 = All ≥ 65 years in HUNT4.

[a]Numbers do not add up to 23,151 (HUNT2), 21,736 (HUNT3), and 14,667 (HUNT4) because of missing information. In HUNT2, the proportion of subjects with abstinence and current drinking does not add up to 100% because the definition of these drinking patterns is based on two different alcohol questions.

[b]Income after taxes. Values of zero or negative income for the year of participation were replaced by the average of the remaining two values (or one value if only a single value was available). An income of 0 for all three years was replaced with missing.

[c]Abstinence: HUNT2 = total abstinence from alcohol; HUNT3 and HUNT4 = never consumed alcohol or not consumed alcohol during the last year; [d] Current drinking: Consumed alcohol ≥ once a month (HUNT2) or ≥ few times a year (HUNT3, HUNT4); [e] Risky drinking: Drinking ≥8 units of alcohol a week; [f] Heavy episodic drinking: Drinking ≥5 (HUNT3) or ≥6 (HUNT4) units of alcohol in one sitting ≥ once a month.

**Table 3.** Trends in alcohol consumption assessed with self-report among middle-aged and older adults in HUNT2 1995–1997, HUNT3 2006–2008, and HUNT4 2017–2019. Subjects participating in HUNT2 and HUNT3 and/or HUNT4 included.

| | Women | | Men | | Men vs. Women | |
|---|---|---|---|---|---|---|
| | Odds (95% CI) | | Odds (95% CI) | | Odds ratio (95% CI) | p-value |
| **_Abstinence a_** | | | | | | |
| **HUNT2** | | | | | | |
| 43–53/ 54–64/ 65–75 | 0.10 (0.09; 0.11) | | 0.06 (0.05; 0.06) | | 0.56 (0.48; 0.65) | <0.001 |
| 54–64/ 65–75/ 76–86 | 0.21 (0.19; 0.23) | | 0.08 (0.07; 0.09) | | 0.41 (0.35; 0.47) | <0.001 |
| ≥65/ ≥76/ ≥87 | 0.43 (0.39; 0.47) | | 0.14 (0.12; 0.16) | | 0.32 (0.27; 0.38) | <0.001 |
| **HUNT3** | | | | | | |
| 43–53/ 54–64/ 65–75 | 0.11 (0.10; 0.12) | | 0.06 (0.05; 0.07) | | 0.53 (0.46; 0.62) | <0.001 |
| 54–64/ 65–75/ 76–86 | 0.23 (0.21; 0.24) | | 0.10 (0.09; 0.11) | | 0.44 (0.38; 0.51) | <0.001 |
| ≥65/ ≥76/ ≥87 | 0.51 (0.47; 0.56) | | 0.21 (0.19; 0.24) | | 0.42 (0.36; 0.49) | <0.001 |
| **HUNT4** | | | | | | |
| 43–53/ 54–64/ 65–75 | 0.22 (0.20; 0.23) | | 0.10 (0.09; 0.12) | | 0.48 (0.42; 0.54) | <0.001 |
| 54–64/ 65–75/ 76–86 | 0.55 (0.51; 0.60) | | 0.21 (0.19; 0.24) | | 0.39 (0.33; 0.45) | <0.001 |
| ≥65/ ≥76/ ≥87 | 1.39 (1.14; 1.64) | | 0.61 (0.47; 0.74) | | 0.44 (0.33; 0.58) | <0.001 |
| **HUNT3 vs. HUNT2** | | | | | | |
| | Odds ratio (95% CI) | p-value | Odds ratio (95% CI) | p-value | Odds ratio (95% CI) | p-value |
| 43–53/ 54–64/ 65–75 | 1.09 (0.96; 1.23) | 0.173 | 1.04 (0.88; 1.23) | 0.650 | 1 | 0.385 |
| 54–64/ 65–75/ 76–86 | 1.08 (0.96; 1.22) | 0.187 | 1.18 (0.99; 1.40) | 0.062 | 1.14 (0.85; 1.54) | 0.059 |
| ≥65/ ≥76/ ≥87 | 1.20 (1.05; 1.37) | 0.006 | 1.55 (1.28; 1.88) | <0.001 | 1.35 (0.99; 1.85) | |
| **HUNT4 vs. HUNT2** | | | | | | |
| | Odds ratio (95% CI) | p-value | Odds ratio (95% CI) | p-value | Odds ratio (95% CI) | p-value |
| 43–53/ 54–64/ 65–75 | 2.16 (1.93; 2.42) | <0.001 | 1.86 (1.59; 2.17) | <0.001 | 1 | 0.481 |
| 54–64/ 65–75/ 76–86 | 2.67 (2.37; 3.01) | <0.001 | 2.55 (2.14; 3.03) | <0.001 | 1.11 (0.83; 1.47) | **0.023**\*\* |
| ≥65/ ≥76/ ≥87 | 3.25 (2.66; 3.97) | <0.001 | 4.38 (3.35; 5.72) | <0.001 | 1.57 (1.06; 2.30) | |
| **HUNT4 vs. HUNT3** | | | | | | |
| | Odds ratio (95% CI) | p-value | Odds ratio (95% CI) | p-value | Odds ratio (95% CI) | p-value |
| 43–53/ 54–64/ 65–75 | 1.98 (1.77; 2.22) | <0.001 | 1.79 (1.53; 2.09) | <0.001 | 1 | 0.840 |
| 54–64/ 65–75/ 76–86 | 2.46 (2.18; 2.78) | <0.001 | 2.16 (1.82; 2.55) | <0.001 | 0.97 (0.73; 1.29) | 0.454 |
| ≥65/ ≥76/ ≥87 | 2.71 (2.21; 3.31) | <0.001 | 2.82 (2.18; 3.66) | <0.001 | 1.16 (0.79; 1.69) | |
| **_Current drinking b_** | | | | | | |
| **HUNT2** | | | | | | |
| 43–53/ 54–64/ 65–75 | 2.19 (2.06; 2.32) | | 5.50 (5.08; 5.92) | | 2.51 (2.28; 2.76) | <0.001 |
| 54–64/ 65–75/ 76–86 | 1.20 (1.11; 1.29) | | 3.44 (3.14; 3.74) | | 2.86 (2.55; 3.21) | <0.001 |
| ≥65/ ≥76/ ≥87 | 0.56 (0.50; 0.63) | | 1.79 (1.58; 1.99) | | 3.18 (2.71; 3.73) | <0.001 |
| **HUNT3** | | | | | | |
| 43–53/ 54–64/ 65–75 | 9.12 (8.32; 9.92) | | 17.10 (15.02; 19.18) | | 1.87 (1.61; 2.18) | <0.001 |
| 54–64/ 65–75/ 76–86 | 4.44 (4.07; 4.82) | | 10.06 (8.85; 11.27) | | 2.26 (1.95; 2.62) | <0.001 |
| ≥65/ ≥76/ ≥87 | 1.94 (1.76; 2.13) | | 4.65 (4.06; 5.25) | | 2.39 (2.04; 2.80) | <0.001 |
| **HUNT4** | | | | | | |
| 43–53/ 54–64/ 65–75 | 4.60 (4.26; 4.94) | | 9.56 (8.60; 10.52) | | 2.08 (1.84; 2.35) | <0.001 |
| 54–64/ 65–75/ 76–86 | 1.80 (1.65; 1.96) | | 4.66 (4.11; 5.22) | | 2.58 (2.23; 2.99) | <0.001 |
| ≥65/ ≥76/ ≥87 | 0.72 (0.59; 0.85) | | 1.65 (1.28; 2.02) | | 2.29 (1.72; 3.05) | <0.001 |
| **HUNT3 vs. HUNT2** | | | | | | |
| | Odds ratio (95% CI) | p-value | Odds ratio (95% CI) | p-value | Odds ratio (95% CI) | p-value |
| 43–53/ 54–64/ 65–75 | 4.16 (3.74; 4.63) | <0.001 | 3.11 (2.69; 3.59) | <0.001 | 1 | 0.668 |
| 54–64/ 65–75/ 76–86 | 3.70 (3.30; 4.15) | <0.001 | 2.93 (2.52; 3.39) | <0.001 | 1.06 (0.82; 1.37) | 0.954 |
| ≥65/ ≥76/ ≥87 | 3.46 (2.98; 4.00) | <0.001 | 2.60 (2.20; 3.09) | <0.001 | 1.01 (0.76; 1.34) | |

*(Continued)*

| | Women | | Men | | Men vs. Women | |
|---|---|---|---|---|---|---|
| | **Odds (95% CI)** | | **Odds (95% CI)** | | **Odds ratio (95% CI)** | **p-value** |
| **HUNT4 vs. HUNT2** | | | | | | |
| | **Odds ratio (95% CI)** | **p-value** | **Odds ratio (95% CI)** | **p-value** | **Odds ratio (95% CI)** | **p-value** |
| 43–53/ 54–64/ 65–75 | 2.10 (1.91; 2.31) | <0.001 | 1.74 (1.53; 2.97) | <0.001 | 1 | 0.491 |
| 54–64/ 65–75/ 76–86 | 1.50 (1.34; 1.69) | <0.001 | 1.36 (1.17; 1.57) | <0.001 | 1.09 (0.85; 1.39) | 0.460 |
| ≥65/ ≥ 76/ ≥ 87 | 1.28 (1.03; 1.58) | 0.023 | 0.92 (0.72; 1.19) | 0.528 | 0.87 (0.61; 1.25) | |
| **HUNT4 vs. HUNT3** | | | | | | |
| | **Odds ratio (95% CI)** | **p-value** | **Odds ratio (95% CI)** | **p-value** | **Odds ratio (95% CI)** | **p-value** |
| 43–53/ 54–64/ 65–75 | 0.50 (0.45; 0.57) | <0.001 | 0.56 (0.48; 0.65) | <0.001 | 1 | 0.838 |
| 54–64/ 65–75/ 76–86 | 0.41 (0.36; 0.46) | <0.001 | 0.46 (0.39; 0.55) | <0.001 | 1.03 (0.77; 1.37) | 0.454 |
| ≥65/ ≥ 76/ ≥ 87 | 0.37 (0.30; 0.45) | <0.001 | 0.35 (0.27; 0.46) | <0.001 | 0.86 (0.59; 1.27) | |
| ***Risky drinking c*** | | | | | | |
| **HUNT2** | | | | | | |
| 43–53/ 54–64/ 65–75 | 0.01 (0.01; 0.01) | | 0.05 (0.05; 0.06) | | 5.33 (4.01; 7.10) | <0.001 |
| 54–64/ 65–75/ 76–86 | 0.01 (0.00; 0.01) | | 0.05 (0.04; 0.05) | | 6.96 (4.46; 10.85) | <0.001 |
| ≥65/ ≥ 76/ ≥ 87 | 0.00 (0.00; 0.01) | | 0.03 (0.02; 0.04) | | 8.06 (3.61; 17.99) | <0.001 |
| **HUNT3** | | | | | | |
| 43–53/ 54–64/ 65–75 | 0.02 (0.02; 0.03) | | 0.09 (0.08; 0.10) | | 3.90 (3.16; 4.81) | <0.001 |
| 54–64/ 65–75/ 76–86 | 0.02 (0.01; 0.02) | | 0.06 (0.05; 0.07) | | 3.41 (2.51; 4.63) | <0.001 |
| ≥65/ ≥ 76/ ≥ 87 | 0.01 (0.00; 0.01) | | 0.04 (0.03; 0.05) | | 6.62 (3.24; 13.53) | <0.001 |
| **HUNT4** | | | | | | |
| 43–53/ 54–64/ 65–75 | 0.03 (0.02; 0.03) | | 0.10 (0.09; 0.11) | | 3.90 (3.14; 4.85) | <0.001 |
| 54–64/ 65–75/ 76–86 | 0.02 (0.01; 0.02) | | 0.06 (0.05; 0.07) | | 3.81 (2.47; 5.88) | <0.001 |
| ≥65/ ≥ 76/ ≥ 87 | NA* | | 0.03 (0.01; 0.05) | | NA* | |
| **HUNT3 vs. HUNT2** | | | | | | |
| | **Odds ratio (95% CI)** | **p-value** | **Odds ratio (95% CI)** | **p-value** | **Odds ratio (95% CI)** | **p-value** |
| 43–53/ 54–64/ 65–75 | 2.27 (1.65; 3.11) | <0.001 | 1.66 (1.41; 1.94) | <0.001 | 1 | 0.226 |
| 54–64/ 65–75/ 76–86 | 2.74 (1.68; 4.46) | <0.001 | 1.34 (1.07; 1.69) | 0.012 | 0.67 (0.35; 1.28) | 0.840 |
| ≥65/ ≥ 76/ ≥ 87 | 1.58 (0.59; 4.25) | 0.364 | 1.30 (0.85; 1.98) | 0.224 | 1.12 (0.36; 3.49) | |
| **HUNT4 vs. HUNT2** | | | | | | |
| | **Odds ratio (95% CI)** | **p-value** | **Odds ratio (95% CI)** | **p-value** | **Odds ratio (95% CI)** | **p-value** |
| 43–53/ 54–64/ 65–75 | 2.63 (1.91; 3.63) | <0.001 | 1.93 (1.64; 2.26) | <0.001 | 1 | 0.428 |
| 54–64/ 65–75/ 76–86 | 2.48 (1.42; 4.34) | 0.001 | 1.36 (1.04; 1.78) | 0.026 | 0.75 (0.37; 1.53) | |
| ≥65/ ≥ 76/ ≥ 87 | NA* | | 1.12 (0.50; 2.51) | 0.791 | NA* | |
| **HUNT4 vs. HUNT3** | | | | | | |
| | **Odds ratio (95% CI)** | **p-value** | **Odds ratio (95% CI)** | **p-value** | **Odds ratio (95% CI)** | **p-value** |
| 43–53/ 54–64/ 65–75 | 1.16 (0.89; 1.51) | 0.269 | 1.16 (1.00; 1.35) | 0.046 | 1 | 0.727 |
| 54–64/ 65–75/ 76–86 | 0.91 (0.57; 1.44) | 0.676 | 1.01 (0.78; 1.31) | 0.934 | 1.12 (0.61; 2.05) | |
| ≥65/ ≥ 76/ ≥ 87 | NA* | | 0.86 (0.38; 1.92) | 0.713 | NA* | |
| ***Heavy episodic drinking d*** | | | | | | |
| **HUNT3** | | | | | | |
| 43–53/ 54–64/ 65–75 | 0.09 (0.09; 0.10) | | 0.49 (0.46; 0.52) | | 5.15 (4.61; 5.76) | <0.001 |
| 54–64/ 65–75/ 76–86 | 0.04 (0.03; 0.05) | | 0.20 (0.18; 0.22) | | 4.84 (3.98; 5.88) | <0.001 |
| ≥65/ ≥ 76/ ≥ 87 | 0.02 (0.01; 0.02) | | 0.09 (0.07; 0.10) | | 4.69 (3.19; 6.90) | <0.001 |
| **HUNT4** | | | | | | |
| 43–53/ 54–64/ 65–75 | 0.03 (0.02; 0.03) | | 0.09 (0.08; 0.10) | | 3.55 (2.88; 4.37) | <0.001 |
| 54–64/ 65–75/ 76–86 | 0.02 (0.01; 0.02) | | 0.04 (0.03; 0.04) | | 2.00 (1.32; 3.03) | 0.001 |
| ≥65/ ≥ 76/ ≥ 87 | 0.01 (0.00; 0.02) | | 0.01 (0.00; 0.02) | | 0.57 (0.11; 2.96) | 0.504 |

*(Continued)*

**Table 3.** (Continued)

| | Women | | Men | | Men vs. Women | |
|---|---|---|---|---|---|---|
| | **Odds (95% CI)** | | **Odds (95% CI)** | | **Odds ratio (95% CI)** | **p-value** |
| **HUNT4 vs. HUNT3** | | | | | | |
| | **Odds ratio (95% CI)** | **p-value** | **Odds ratio (95% CI)** | **p-value** | **Odds ratio (95% CI)** | **p-value** |
| 43–53/ 54–64/ 65–75 | 0.28 (0.23; 0.34) | <0.001 | 0.19 (0.17; 0.22) | <0.001 | 1 | 0.052 |
| 54–64/ 65–75/ 76–86 | 0.43 (0.30; 0.63) | <0.001 | 0.18 (0.14; 0.23) | <0.001 | 0.60 (0.36; 1.00) | **0.047**** |
| ≥65/ ≥ 76/ ≥ 87 | 0.63 (0.25; 1.63) | 0.342 | 0.08 (0.02; 0.31) | <0.001 | 0.18 (0.03; 0.97) | |

Abbreviations: CI = confidence interval; HUNT = Trøndelag Health Study.

[a]Abstinence: HUNT2 = total abstinence from alcohol; HUNT3 and HUNT4 = never consumed alcohol or not consumed alcohol during the last year; [b]Current drinking: Consumed alcohol ≥ once a month (HUNT2) or ≥ few times a year (HUNT3, HUNT4); [c]Risky drinking: Drinking ≥8 units of alcohol a week; [d]Heavy episodic drinking: Drinking ≥5 (HUNT3) or ≥6 (HUNT4) units of alcohol in one sitting ≥ once a month.

*NA = not available due to low frequencies.

**The p-value indicates significant differences in changes in abstinence and heavy episodic drinking from HUNT2 to HUNT4 between women and men in the age groups ≥65/ ≥ 76/ ≥ 87 years compared to the age groups 43–53/54–64/65–75 years (three-way interaction in a linear mixed model).

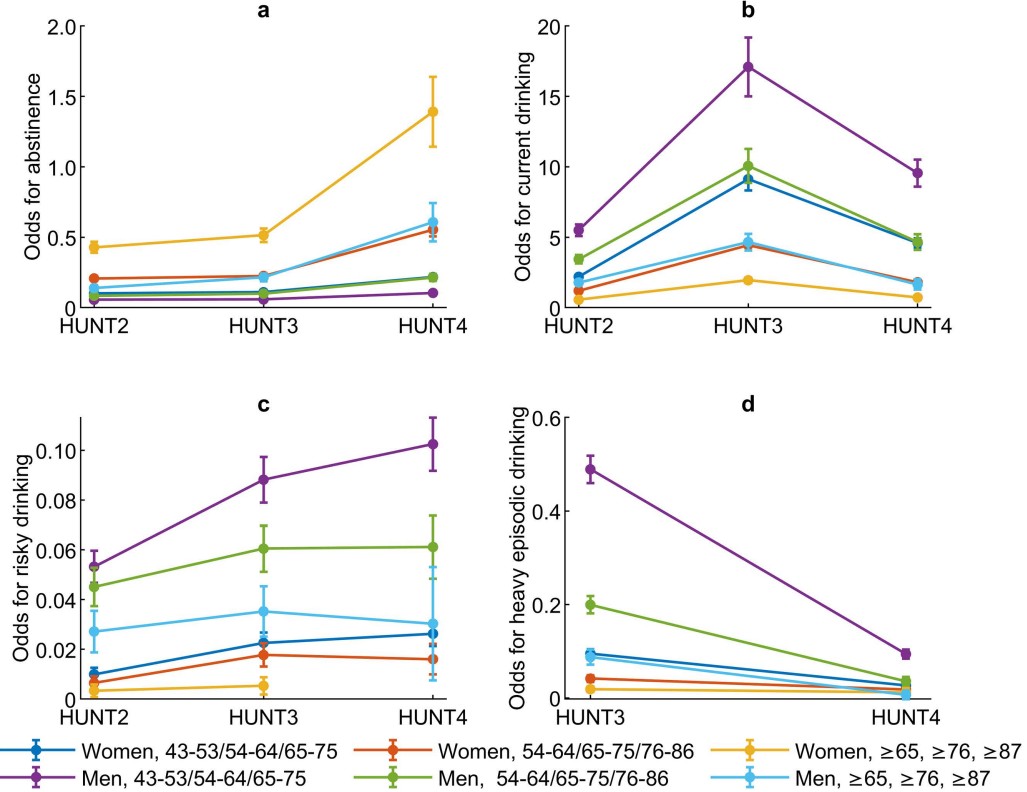

**Fig 2. a-d. Sex- and age-specific differences in trend in self-reported alcohol consumption. Subjects participating in HUNT2 and HUNT3 and/ or HUNT4 included. Illustration of results in Table 3.** Odds for trends in 2a) abstinence; 2b) current drinking; 2c) risky drinking; and 2d) heavy episodic drinking among women and men with increasing age from HUNT2 to HUNT3 and HUNT4.

**Table 4. Change in alcohol consumption assessed with phosphatidylethanol among middle-aged and older adults in HUNT3 2006–2008 and HUNT4 2017–2019. Subjects participating in HUNT2 and HUNT3 and/or HUNT4 included.**

| | Women | | Men | | Men vs. women | |
|---|---|---|---|---|---|---|
| | **Odds (95% CI)** | | **Odds (95% CI)** | | **Odds ratio (95% CI)** | **p-value** |
| **PEth < 0.03 µmol/l** | | | | | | |
| **HUNT3** | | | | | | |
| 43–53/ 54–64/ 65–75 | 1.69 (1.56; 1.82) | | 0.83 (0.77; 0.90) | | 0.49 (0.44; 0.55) | <0.001 |
| 54–64/ 65–75/ 76–86 | 3.14 (2.81; 3.47) | | 1.34 (1.21; 1.47) | | 0.43 (0.37; 0.49) | <0.001 |
| ≥65/ ≥ 76/ ≥ 87 | 8.92 (7.14; 10.69) | | 3.09 (2.62; 3.56) | | 0.35 (0.27; 0.44) | <0.001 |
| **HUNT4** | | | | | | |
| 43–53/ 54–64/ 65–75 | 2.12 (1.92; 2.31) | | 1.18 (1.07; 1.28) | | 0.56 (0.49; 0.63) | <0.001 |
| 54–64/ 65–75/ 76–86 | 6.74 (5.50; 7.99) | | 3.27 (2.74; 3.79) | | 0.48 (0.38; 0.62) | <0.001 |
| ≥65/ ≥ 76/ ≥ 87 | 24.33 (8.11; 40.55) | | 7.76 (3.84; 11.69) | | 0.32 (0.14; 0.74) | 0.007 |
| **HUNT4 vs. HUNT3** | | | | | | |
| | **Odds ratio (95% CI)** | **p-value** | **Odds ratio (95% CI)** | **p-value** | **Odds ratio (95% CI)** | **p-value** |
| 43–53/ 54–64/ 65–75 | 1.25 (1.11; 1.41) | <0.001 | 1.41 (1.26; 1.59) | <0.001 | 1 | 0.969 |
| 54–64/ 65–75/ 76–86 | 2.15 (1.74; 2.66) | <0.001 | 2.44 (2.02; 2.93) | <0.001 | 1.01 (0.73; 1.40) | 0.656 |
| ≥65/ ≥ 76/ ≥ 87 | 2.73 (1.36; 5.47) | 0.005 | 2.52 (1.48; 4.26) | 0.001 | 0.82 (0.34; 1.99) | |
| **PEth 0.03–0.09 µmol/l** | | | | | | |
| **HUNT3** | | | | | | |
| 43–53/ 54–64/ 65–75 | 0.25 (0.23; 0.28) | | 0.37 (0.34; 0.40) | | 1.47 (1.30; 1.66) | <0.001 |
| 54–64/ 65–75/ 76–86 | 0.16 (0.14; 0.18) | | 0.27 (0.24; 0.30) | | 1.66 (1.40; 1.98) | <0.001 |
| ≥65/ ≥ 76/ ≥ 87 | 0.07 (0.06; 0.09) | | 0.15 (0.12; 0.18) | | 2.06 (1.51; 2.79) | <0.001 |
| **HUNT4** | | | | | | |
| 43–53/ 54–64/ 65–75 | 0.24 (0.21; 0.26) | | 0.30 (0.27; 0.33) | | 1.28 (1.10; 1.48) | 0.002 |
| 54–64/ 65–75/ 76–86 | 0.09 (0.07; 0.11) | | 0.15 (0.12; 0.18) | | 1.74 (1.28; 2.35) | <0.001 |
| ≥65/ ≥ 76/ ≥ 87 | 0.04 (0.01; 0.06) | | 0.07 (0.03; 0.12) | | 1.98 (0.76; 5.13) | 0.161 |
| **HUNT4 vs. HUNT3** | | | | | | |
| | **Odds ratio (95% CI)** | **p-value** | **Odds ratio (95% CI)** | **p-value** | **Odds ratio (95% CI)** | **p-value** |
| 43–53/ 54–64/ 65–75 | 0.94 (0.81; 1.08) | 0.347 | 0.81 (0.71; 0.93) | 0.003 | 1 | 0.369 |
| 54–64/ 65–75/ 76–86 | 0.55 (0.43; 0.72) | <0.001 | 0.58 (0.46; 0.73) | <0.001 | 1.20 (0.81; 1.79) | 0.843 |
| ≥65/ ≥ 76/ ≥ 87 | 0.50 80.24; 1.05) | 0.066 | 0.48 (0.24; 0.94) | 0.031 | 1.11 (0.40; 3.08) | |
| **PEth 0.10–0.29 µmol/l** | | | | | | |
| **HUNT3** | | | | | | |
| 43–53/ 54–64/ 65–75 | 0.13 (0.12; 0.15) | | 0.23 (0.21; 0.25) | | 1.78 (1.53; 2.06) | <0.001 |
| 54–64/ 65–75/ 76–86 | 0.07 (0.05; 0.08) | | 0.15 (0.13; 0.17) | | 2.33 (1.85; 2.94) | <0.001 |
| ≥65/ ≥ 76/ ≥ 87 | 0.02 (0.01; 0.03) | | 0.09 (0.07; 0.11) | | 3.61 (2.26; 5.74) | <0.001 |
| **HUNT4** | | | | | | |
| 43–53/ 54–64/ 65–75 | 0.11 (0.09; 0.12) | | 0.19 (0.17; 0.21) | | 1.80 (1.49; 2.17) | <0.001 |
| 54–64/ 65–75/ 76–86 | 0.04 (0.02; 0.05) | | 0.09 (0.07; 0.12) | | 2.56 (1.69; 3.87) | <0.001 |
| ≥65/ ≥ 76/ ≥ 87 | 0.00 (0.00; 0.01) | | 0.03 (0.00; 0.07) | | 7.88 (0.91; 68.12) | 0.061 |
| **HUNT4 vs. HUNT3** | | | | | | |
| | **Odds ratio (95% CI)** | **p-value** | **Odds ratio (95% CI)** | **p-value** | **Odds ratio (95% CI)** | **p-value** |
| 43–53/ 54–64/ 65–75 | 0.80 (0.67; 0.97) | 0.019 | 0.81 (0.70; 0.95) | 0.010 | 1 | 0.765 |
| 54–64/ 65–75/ 76–86 | 0.55 (0.38; 0.82) | 0.003 | 0.61 (0.46; 0.81) | 0.001 | 1.08 (0.64; 1.85) | 0.497 |
| ≥65/ ≥ 76/ ≥ 87 | 0.18 (0.02; 1.37) | 0.098 | 0.40 (0.16; 1.02) | 0.054 | 2.16 (0.23; 19.87) | |
| **PEth ≥ 0.30 µmol/l** | | | | | | |
| **HUNT3** | | | | | | |
| 43–53/ 54–64/ 65–75 | 0.06 (0.05; 0.07) | | 0.09 (0.08; 0.11) | | 1.64 (1.33; 2.02) | <0.001 |
| 54–64/ 65–75/ 76–86 | 0.04 (0.03; 0.05) | | 0.09 (0.08; 0.11) | | 2.11 (1.59; 2.78) | <0.001 |
| ≥65/ ≥ 76/ ≥ 87 | 0.01 (0.00; 0.02) | | 0.04 (0.02; 0.05) | | 3.81 (1.86; 7.81) | <0.001 |

*(Continued)*

Table 4. (Continued)

| | Women | | Men | | Men vs. women | |
|---|---|---|---|---|---|---|
| | Odds (95% CI) | | Odds (95% CI) | | Odds ratio (95% CI) | p-value |
| **HUNT4** | | | | | | |
| 43–53/ 54–64/ 65–75 | 0.04 (0.03; 0.04) | | 0.07 (0.06; 0.09) | | 2.06 (1.54; 2.75) | <0.001 |
| 54–64/ 65–75/ 76–86 | 0.01 (0.01; 0.02) | | 0.02 (0.01; 0.03) | | 1.29 (0.60; 2.76) | 0.509 |
| ≥65/ ≥ 76/ ≥ 87 | NA* | | 0.01 (−0.01; 0.03) | | NA* | |
| **HUNT4 vs. HUNT3** | | | | | | |
| | Odds ratio (95% CI) | p-value | Odds ratio (95% CI) | p-value | Odds ratio (95% CI) | p-value |
| 43–53/ 54–64/ 65–75 | 0.62 (0.47; 0.83) | 0.001 | 0.78 (0.63; 0.98) | 0.031 | 1 | 0.112 |
| 54–64/ 65–75/ 76–86 | 0.30 (0.16; 0.54) | <0.001 | 0.18 (0.10; 0.32) | <0.001 | 0.49 (0.20; 1.18) | |
| ≥65/ ≥ 76/ ≥ 87 | NA* | | 0.38 (0.09; 1.60) | 0.187 | NA* | |

Abbreviations: CI = confidence interval; HUNT = Trøndelag Health Study; PEth = phosphatidylethanol 16:0/18:1.

*NA = not available due to low frequencies.

## Prevalence and change in alcohol consumption measured with PEth

The proportion of those with PEth concentrations <0.03 µmol/l increased among women and men from 71.9% and 54.4% in HUNT3 (≥54 years) to 75.5% and 62.3% in HUNT4 (≥65 years), respectively. The prevalence of all other categorized PEth concentrations (PEth 0.03–0.09, 0.10–0.29, and ≥0.30 µmol/l) decreased among both women and men from HUNT3 to HUNT4 (Table 2).

Among both women and men, the increase in PEth concentrations <0.03 µmol/l from HUNT3 to HUNT4 was significant in all age groups, whereas the decrease in PEth concentrations of 0.03–0.09, 0.10–0.29, and ≥0.30 µmol/l was significant in most age groups (Table 4 and Fig 3). It was not sufficient data available in the oldest age group of women in HUNT4 to perform the analyses for PEth concentrations ≥0.30 µmol/l.

## Sex- and age-specific differences in alcohol consumption measured with PEth

Women had PEth concentrations <0.03 µmol/l more often than men in all age groups in HUNT3 and HUNT4 whereas men were more likely to have PEth concentrations ≥0.03 µmol/l (PEth 0.03–0.09, 0.10–0.29, and ≥0.30 µmol/l) than women in all age groups in HUNT3 and in the youngest age group in HUNT4 (Table 4).

There were no significant sex- or age-specific differences in the change in the categories of PEth concentrations from HUNT3 to HUNT4 (Table 4).

## Discussion

In this large population-based longitudinal HUNT Study, we examined trends in self-reported alcohol consumption and alcohol consumption measured with PEth among women and men from early mid-age (≥43 years) to older age (≥65 years) by using data from the three most recent HUNT surveys (HUNT2 1995−1997, HUNT3 2006−2008, and HUNT4 2017−2019). We found that the proportion of subjects with abstinence from alcohol and PEth <0.03 µmol/l increased from HUNT2 and/or HUNT3 to HUNT4 whereas heavy episodic drinking and categorized PEth concentrations ≥0.03 µmol/l (0.03–0.09, 0.10–0.29, and ≥0.30 µmol/l) decreased from HUNT3 to HUNT4 in both women and men in most age groups. The trend in risky drinking increased from HUNT2 to HUNT3 and from HUNT2 to HUNT4 in women and men in the two youngest age groups but only among men in the youngest age group from HUNT3 to HUNT4. Men were more likely to consume alcohol than women assessed with both self-report and with PEth in most age groups. In the oldest age group, a convergence between the sexes regarding abstinence and heavy episodic drinking was observed which was mostly caused by changes in men.

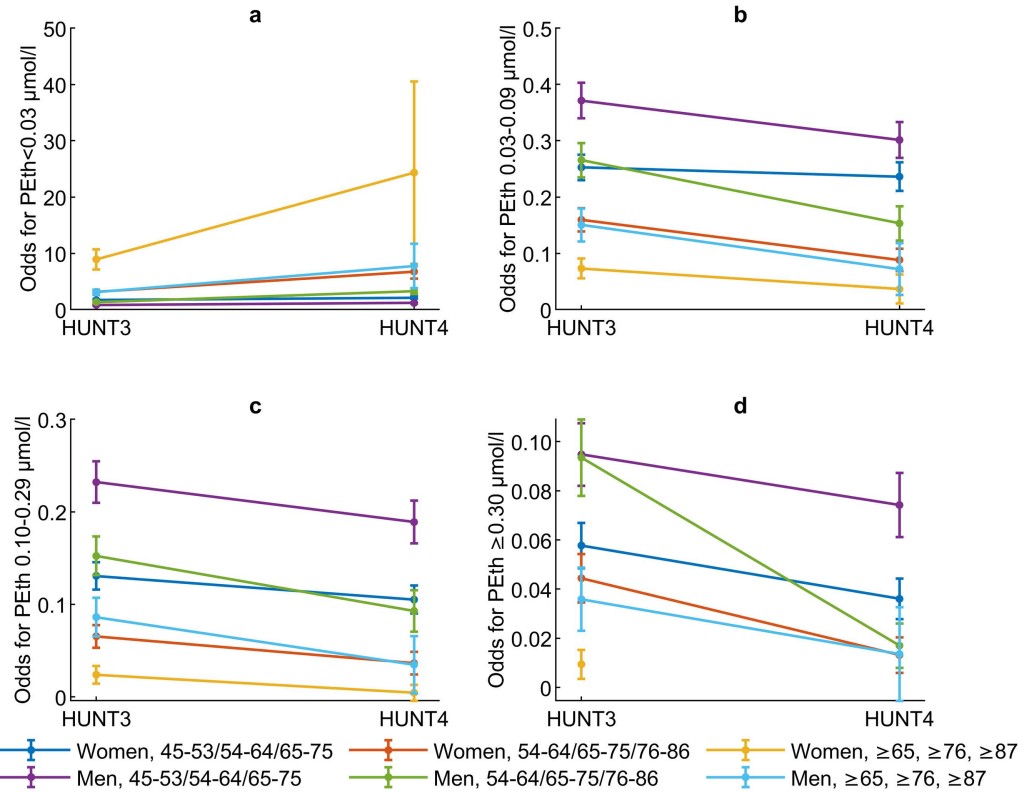

**Fig 3. a-d. Sex- and age-specific differences in change in alcohol consumption measured with phosphatidylethanol 16:0/18:1. Subjects participating in HUNT2 and HUNT3 and/or HUNT4 included.** Odds for changes in 3a) <0.03 µmol/l; 3b) 0.03–0.09 µmol/l; 3c) 0.10–0.29 µmol/l; and 3d) ≥0.30 µmol/l in women and men with increasing age.

## Trends and sex differences in alcohol consumption

A main finding in our study was the general trend towards a less heavy drinking pattern with age in the period from HUNT3 to HUNT4. In women and men of all age categories, except of the oldest women, the heavy episodic drinking and the PEth concentrations indicating high volume intake (0.10–0.29 µmol/l and ≥0.30 µmol/l) [38] were reduced. A decline in heavy episodic drinking is in line with previous longitudinal studies conducted in Ireland [64] and the US [65] which concluded with an overall decline in heavy episodic drinking as the individuals aged from middle age to older age. However, our finding was not in line with another US longitudinal study [4], which found that heavy episodic drinking, defined as ≥5 drinks in a single day in the past year, was stable among men (≥60 years at baseline) during a 17 year follow up period, while it increased on average 3.7% per year among women (≥60 years at baseline). Varying definitions of heavy episodic drinking complicate the comparison of the results both with our study and other studies [4,64]. However, differences in heavy episodic drinking might also be due to variations in drinking cultures and social norms between Western countries.

Furthermore, women were less likely to engage in heavy episodic drinking and have PEth concentrations ≥0.03 µmol/l than men, but among the middle-aged or the oldest age group in HUNT4, women and men were just as likely to self-report heavy episodic drinking or have PEth concentrations ≥0.03 µmol/l. These results are somewhat unexpected as the traditional sex differences in alcohol consumption have been that men consume more alcohol than women [6,29,31]. We also found a convergence between the sexes in heavy episodic drinking from HUNT3 to HUNT4, as the change towards less heavy episodic drinking was more pronounced in men than women. Convergence between the sexes in alcohol

consumption has also been found in previous research [4,29,66] and might be of concern due to women's higher sensitivity to alcohol consumption compared to men [32–34].

Another important finding was the increasing trend in abstinence among both women and men with increasing age from HUNT2 and HUNT3 to HUNT4, which is in line with a previous study conducted in the US [65]. However, the increasing trend in abstinence with increasing age was more pronounced among women than men from HUNT2 to HUNT4. This finding is partly compatible with an Irish longitudinal study including individuals aged 50 years or older at baseline which found that the odds of abstinence increased among women but not among men as they aged [64]. Additionally, we observed that PEth levels <0.03 µmol/l increased among women and men from HUNT3 to HUNT4. However, since PEth concentrations <0.03 µmol/l indicate abstinence or low levels of alcohol consumption [54], the increase in the prevalence of PEth <0.03 µmol/l is not directly comparable to the increasing trend in self-reported abstinence. According to the sex difference in abstinence or very low level of alcohol consumption, we found, consistent with previous research [6,29,67], that women were more likely to abstain from alcohol and have PEth concentrations <0.03 µmol/l than men. However, we found a convergence between the sexes in abstinence from HUNT2 to HUNT4.

In the present study we lacked information on the causes of the overall decline and convergence between the sexes in alcohol consumption with age. However, previous research has identified health precautions as the most commonly reported reason for reducing alcohol intake or stop drinking in older age [19,64,68]. This phenomenon is also known as the 'sick quitter' effect [19]. Other factors contributing to a decline in alcohol consumption in later life include the loss of a partner [21], lower income [21], and fewer social opportunities to drink alcohol [19]. The decline in alcohol consumption with increasing age might also be due to selective mortality, where heavy drinkers die earlier, while lighter drinkers survive and generally continue their lighter drinking patterns as they age [21]. The convergence between the sexes in alcohol consumption among middle-aged and older adults might be explained by cultural changes, increased social acceptance of alcohol consumption among women, greater equality between the sexes, and women's entry into the labor market which has provided them with more financial independence [43,69,70].

Even though the overall trend in our study indicated less heavy drinking patterns as measured with both heavy episodic drinking and PEth ≥0.10 µmol/l [38], we observed an increasing trend in risky drinking among women and men in the two youngest age groups from HUNT2 to HUNT3 and from HUNT2 to HUNT4, but only among men in the youngest age group from HUNT3 to HUNT4. As expected, and in line with previous research [29,67,71], we found that women were less likely to have a risky drinking pattern than men. The increasing trend in risky drinking among both women and men may be attributed to more positive attitudes towards alcohol consumption over the past decades, as well as higher levels of education, improved standards of living, increased leisure time, and better overall health [6,29,72]. Additionally, an active lifestyle, characterized by higher levels of social interactions and vacations in countries with a Mediterranean drinking culture, may have contributed to this increasing trend [19,73–75]. Our finding was not consistent with two studies reporting a decline in a risky drinking pattern in both sexes as they aged from middle to older age [76,77]. However, other studies have found that some individuals seem to develop a risky drinking pattern later in life [22,23,67]. A Spanish study in older community dwelling adults (>60 years) reported that about 9% of women and 23% of men transitioned from a light drinking pattern to a risky drinking pattern during the study period [67]. The increasing trend in risky drinking observed in our study may increase the risk of alcohol-related health problems in an ageing population in the future [78,79]. However, the increasing trend in risky drinking was not accompanied by a corresponding increase in heavy episodic drinking or PEth values ≥0.10 µmol/l. Even so, since we defined risky drinking as consuming ≥8 units of alcohol per week, individuals can most likely drink 1–2 units of alcohol per day, especially if taken with food, without reaching PEth concentrations >0.10 µmol/l [35,38].

Furthermore, risky drinking among older adults (≥65 years) in HUNT4 was quite infrequent (5.1%) and considerably lower compared to a European survey of those aged 60–79 years, which found a prevalence of risky drinking (≥8 drinks/week) of 42% in Denmark, 36% in Belgium, and 23% in Portugal [80]. Also, Norwegian older adults showed a higher prevalence of risky drinking in that study (26%) [80] than in our study.

## Clinical implications

Alcohol consumption is an important preventable cause of global morbidity and mortality [2,3,81]. Even at low level, alcohol consumption may be more harmful in older adults than in middle-aged and younger adults as the concentrations of alcohol will be higher at a given intake and its effects may be exacerbated due to multimorbidity, polypharmacy, and interaction between alcohol and medications [4,10,24,82]. Thus, although the present study found a trend toward increased proportion of participants being abstinent, the concurrent rise in risky drinking among both sexes, may increase the demand on the Norwegian healthcare resource allocation due to the elevated risk of alcohol-related conditions such as falls, injuries, adverse medication interactions, and chronic diseases [3]. Women may be of particular risk of the adverse events to alcohol consumption [32–34]. Similar trends in increased risky drinking among middle-aged and older adults have been observed in other countries [24,71,78] underscoring similar public health implications.

It is recommended that health care professionals in primary health care as well as in the specialist health services incorporate assessment of alcohol consumption in the clinical management of middle-aged and older adults [83–86]. Alcohol assessment can be performed using pragmatic case finding [83,85] or screening tools such as AUDIT (Alcohol Use Disorders Identification Test) [87] or CAGE (Cut down, Annoyed, Guilty, and Eye-opener) [88]. Some general practitioners have advocated routinely use of PEth in primary health care to identify risky or harmful alcohol consumption [89,90] but ethical principles, including consent from the patient, should guide such use [90,91]. Given that older adults with risky or harmful drinking patterns may still be active drivers, general practitioners could consider alcohol screening questionnaires and the use of PEth during the judgments of driver's license renewals [16,92]. They could also promote alternatives for safe transportation, especially in rural areas [16].

For individuals identified with a risky or a harmful drinking pattern, a brief intervention has been shown to be effective in the clinical setting [93–95]. Middle-aged individuals prioritized for alcohol intervention may thus have reduced risk for medication interactions complications and better health in older age than otherwise [24,76,96]. Health authorities could integrate alcohol prevention efforts or campaigns into general practice, senior centers, voluntary organizations or through social media and print media targeting older adults.

## Strengths and limitations

The strengths of the present study include the longitudinal design covering a follow-up period from 12 to 24 years and the high number of participants recruited from the total population in the Nord-Trøndelag region in Central Norway [45,49,50]. The population of Nord-Trøndelag has a homogenous and stable population with little migration [28]. This increases the likelihood that the estimates regarding trends in alcohol consumption and comparisons between sexes and age groups are reliable [28]. The population of Nord-Trøndelag is quite similar to the general population of Norway with some exceptions including the lack of large cities [45,46,49]. Thus, the generalization should be done with caution since inhabitants in larger cities may drink more frequently and more heavily than those in rural areas and smaller cities [29,30]. Even so, the general adult population of Norway (18–79 years) have an alcohol intake comparable with the adult HUNT population [97]. A major strength is the use of PEth as a biological marker of alcohol intake. The use of PEth reduces the potential for the underestimation of alcohol consumption associated with self-report [98].

However, the study has some limitations to consider. At the same time as the migration is low, the ethnical diversity in the population of Nord-Trøndelag is limited, and the findings may not be representative for more ethnically diverse populations of middle-aged and older adults [99]. Furthermore, the presented trends of alcohol consumption are at a group level, and individual differences in the change in alcohol consumption with increasing age are not fully captured in the study. Moreover, the participation rate in the HUNT surveys declined from HUNT2 to HUNT3 but was stable between HUNT3 and HUNT4. The number of participants were lowest in the oldest age groups (≥80 years) due to poor self-rated health and chronic disorders and is also lower in men than women [45,46,49,50]. Furthermore, it is likely that those abstaining and having heavy drinking may be underrepresented, as found in other population-based studies [100,101].

Thus, our findings may be influenced by non-response and selection biases, as the oldest individuals – particularly abstainers or heavy drinkers – may have declined participation due to poor health, resulting in a biased sample toward healthier individuals [102]. These conditions may affect the generalizability of our results, but, given the few epidemiological alcohol studies including those over 70 years, this study contributes important evidence on changes in alcohol consumption into very old age [29].

As the alcohol questions have varied somewhat between the HUNT surveys, it is necessary to interpret the trends in abstinence, current drinking, and heavy episodic drinking with caution [52]. In HUNT2, abstinence and current drinking were defined based on two different alcohol questions (one question about abstinence and one question about drinking frequency last 14 days). For that reason, the total percentage of participants with abstinence and current drinking did not add up to 100% in HUNT2. However, the question regarding the number of units of alcohol (wine, beer, and liquor) usually consumed the previous 14 days has remained the same. Unfortunately, heavy episodic drinking and the PEth concentrations were only assessed in HUNT3 and HUNT4, and the lack of data of risky drinking and PEth concentrations ≥0.30 µmol/l in the oldest women, limits the opportunity to draw conclusions about the trends and change in this group.

Currently, there is no gold standard for self-reported measures of risky drinking or heavy episodic drinking among adults or older adults [60,103,104]. These drinking patterns have been defined in various ways in previous epidemiologic studies, and also differently between women and men [60]. The variability in definitions complicates the comparison of prevalences, changes, and trends in alcohol consumption between studies. A standardized definition of risky drinking and heavy episodic drinking for both the general adult population and for the older population is clearly needed [60,103,104].

Compared to sales data [37,105], self-reported use of alcohol is substantially lower, and most likely under-reported due to factor such as reporting bias and recall bias because of cognitive impairment, memory errors, social desirability bias, and/or stigma associated with alcohol consumption [72,106,107]. Gender role-related expectations and taboos surrounding substance use may contribute to sex-specific differences in the risk of misreporting alcohol consumption [108]. Under-reporting of alcohol consumption might have led to misclassification and underestimation of risky drinking and heavy episodic drinking in our study.

Underestimating of self-reported alcohol consumption, especially among heavy drinkers, [36,37], also complicates the comparison of self-reported alcohol consumption with PEth [56]. PEth is not susceptible to the inherent biases associated with self-reported alcohol consumption as PEth measures the actual ethanol intake during the last 2–4 weeks [62]. Thus, PEth may more accurately identify middle-aged and older adults with risky or harmful drinking than self-reported use [56,109,110]. However, more liberal attitudes and less stigma toward alcohol consumption over the years may contribute to less underreporting [29]. Even so, those abstaining from alcohol and those with heavy drinking may be underrepresented in the present population-based study, as found in other studies [100,101]. Furthermore, longitudinal population-based studies examining the trends in alcohol consumption are also at risk of selection bias because heavier drinkers are more likely to drop out during the follow-up due to poor health or premature death [20].

## Conclusions

With the use of longitudinal data from the three most recent HUNT surveys in Central Norway, we found a trend towards more abstinence and less heavy episodic drinking with increasing age in middle-aged and older adults since the 1990s. Moreover, we found a change towards fewer subjects with PEth concentrations ≥0.10 µmol/l among both women and men as they aged from HUNT3 to HUNT4. However, the prevalence of risky drinking increased from HUNT2 to HUNT4 in both sexes in the two youngest age groups. We observed convergence between the sexes in abstinence and heavy episodic drinking with increasing age. This effect was predominantly caused by a reduced alcohol intake in men. The fact that women reduced their alcohol consumption to a lower degree than men as they aged, might be of concern, as women are more sensitive to alcohol than men.

We did not have information about factors causing the observed changes in alcohol consumption, and this need to be further examined in a Norwegian context.

## Supporting information

**S1 Table. Changes in questions used to define abstinence, current drinking, and heavy episodic drinking from HUNT2 (1995–1997) to HUNT3 (2006–2008) and HUNT4 (2017–2019).**
(DOCX)

## Acknowledgments

We would like to acknowledge the HUNT Study participants and the Trøndelag Health Study (HUNT) which is a collaboration between HUNT Research Centre (Faculty of Medicine and Health Sciences, Norwegian University of Science and Technology, NTNU), Trøndelag County Council, Central Norway Regional Health Authority, and the Norwegian Institute of Public Health. We would also acknowledge the clinical staff at the Department of Clinical Pharmacology at St. Olavs University Hospital in Trondheim (Norway).

## Author contributions

**Formal analysis:** Kjerstin Tevik, Jūratė Šaltytė Benth.

**Methodology:** Kjerstin Tevik, Ragnhild Bergene Skråstad, Jūratė Šaltytė Benth, Geir Selbæk, Sverre Bergh, Olav Spigset, Steinar Krokstad, Anne-Sofie Helvik.

**Writing – original draft:** Kjerstin Tevik, Ragnhild Bergene Skråstad, Jūratė Šaltytė Benth, Geir Selbæk, Sverre Bergh, Olav Spigset, Steinar Krokstad, Anne-Sofie Helvik.

**Writing – review & editing:** Kjerstin Tevik, Ragnhild Bergene Skråstad, Jūratė Šaltytė Benth, Geir Selbæk, Sverre Bergh, Olav Spigset, Steinar Krokstad, Anne-Sofie Helvik.

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
