## [Decision Letter · Decision Letter 0]

3 Sep 2025

PONE-D-25-31670Trends in alcohol consumption in middle-aged and older adults, assessed with self-report and the alcohol marker phosphatidylethanol - A longitudinal HUNT StudyPLOS ONE

Dear Dr. Tevik,

Thank you for submitting your manuscript to PLOS ONE. After careful consideration, we feel that it has merit but does not fully meet PLOS ONE’s publication criteria as it currently stands. Therefore, we invite you to submit a revised version of the manuscript that addresses the points raised during the review process.

We look forward to receiving your revised manuscript.

Kind regards,

Y-h. Taguchi, Dr. Sci.

Academic Editor

PLOS ONE

**Journal Requirements:**

1. When submitting your revision, we need you to address these additional requirements. Please ensure that your manuscript meets PLOS ONE's style requirements, including those for file naming. The PLOS ONE style templates can be found at https://journals.plos.org/plosone/s/file?id=wjVg/PLOSOne_formatting_sample_main_body.pdf and https://journals.plos.org/plosone/s/file?id=ba62/PLOSOne_formatting_sample_title_authors_affiliations.pdf 2. If the reviewer comments include a recommendation to cite specific previously published works, please review and evaluate these publications to determine whether they are relevant and should be cited. There is no requirement to cite these works unless the editor has indicated otherwise. 

**Additional Editor Comments:**

Please revise the manuscript to address the concerns raised by reviewers.

Reviewers' comments:

Reviewer's Responses to Questions

**Comments to the Author**

1. Is the manuscript technically sound, and do the data support the conclusions?

Reviewer #1: Yes

Reviewer #2: Yes

2. Has the statistical analysis been performed appropriately and rigorously? 

Reviewer #1: Yes

Reviewer #2: Yes

3. Have the authors made all data underlying the findings in their manuscript fully available?

Reviewer #1: Yes

Reviewer #2: Yes

4. Is the manuscript presented in an intelligible fashion and written in standard English?

Reviewer #1: Yes

Reviewer #2: Yes

5. Review Comments to the Author

**Reviewer #1: ** Thank you for the invitation to review this article, dear editor. The manuscript provides useful insights into alcohol consumption trends in Norway and makes a valuable contribution. At the same time, a few clarifications and additions could help strengthen the paper and make it more suitable for PLOS ONE readers.

[1] The manuscript notes slight differences in how abstinence and heavy episodic drinking were measured across HUNT2, HUNT3, and HUNT4. The authors could provide a brief table or appendix explicitly summarizing these definitional shifts to aid reader interpretation.

[2] The discussion could more directly tie findings to implications for public health policy in Norway and other countries. For example, what do trends in abstinence and risky drinking among older adults imply for healthcare resource allocation, prevention campaigns, or even some other external problems such as traffic safety interventions?

[3] While the introduction discusses health-related harms of alcohol, it overlooks the well-documented connection between alcohol consumption and external harms such as risky driving on the road. Briefly Including this dimension of risk to other people would strengthen the framing of alcohol as a broader societal risk factor. Consider including these recent literatures, such as https://doi.org/10.1016/j.ijtst.2024.11.009, and https://doi.org/10.1080/23249935.2025.2516817 to strengthen the societal relevance of your work.

[4] The limitations of the paper, such as with the data and methodology and some other factors should be well acknowledged.

**Reviewer #2:**  Strengths

1. The HUNT study is well explained, with survey waves, dates, and participation rates clearly documented.

2. The introduction is well written and provides clear context for the study.

3. The manuscript acknowledges both strengths (generalizability to Norway/Western health trends) and weaknesses (lack of large cities, fewer immigrants, lower education/income) of the Nord-Trøndelag population.

4. Use of questionnaires, registry data (Statistics Norway), and biobank samples is clearly stated.

5. Self-reports complemented with PEth biomarkers strengthens validity.

Minor issues

Indicate the type of model (logistic, multinomial, or linear) and link function should be stated.

Acknowledgement of potential biases (selection, multiple testing)

6. PLOS authors have the option to publish the peer review history of their article (what does this mean? ). If published, this will include your full peer review and any attached files.

**Do you want your identity to be public for this peer review?** For information about this choice, including consent withdrawal, please see our Privacy Policy .

Reviewer #1: No

Reviewer #2: No

---

## [Author Response · Author response to Decision Letter 1]

24 Sep 2025

PONE-D-25-31670

Trends in alcohol consumption in middle-aged and older adults, assessed with self-report and the alcohol marker phosphatidylethanol - A longitudinal HUNT Study

5. Review Comments to the Author

Please use the space provided to explain your answers to the questions above. You may also include additional comments for the author, including concerns about dual publication, research ethics, or publication ethics. (Please upload your review as an attachment if it exceeds 20,000 characters).

We would like to thank the reviewers for their valuable contribution to improving the manuscript. For references to page and line numbers in our responses, please see the Revised Manuscript with Track Changes.

Reviewer #1: Thank you for the invitation to review this article, dear editor. The manuscript provides useful insights into alcohol consumption trends in Norway and makes a valuable contribution. At the same time, a few clarifications and additions could help strengthen the paper and make it more suitable for PLOS ONE readers.

[1] The manuscript notes slight differences in how abstinence and heavy episodic drinking were measured across HUNT2, HUNT3, and HUNT4. The authors could provide a brief table or appendix explicitly summarizing these definitional shifts to aid reader interpretation.

Thank you for this suggestion. We have created a table illustrating how the questions and definitions of abstinence, current drinking, and heavy episodic drinking have changed from HUNT2 to HUNT3 and HUNT4. We included the questions and definitions of current drinking, as these have also changed across the HUNT surveys. The table has been added as an additional file (see S1 Table), and the following sentence is added in lines 205-207 on page 10:

S1 Table describes in detail how the questions and definitions regarding abstinence, current drinking, and heavy episodic drinking have changed from HUNT2 to HUNT3 and HUNT4.

[2] The discussion could more directly tie findings to implications for public health policy in Norway and other countries. For example, what do trends in abstinence and risky drinking among older adults imply for healthcare resource allocation, prevention campaigns, or even some other external problems such as traffic safety interventions?

We appreciate these valuable suggestions. The following text has been added to the section about Clinical implications in the Discussion.

Please read lines 547-551 on page 37:

Thus, although the present study found a trend toward increased proportion of participants being abstinent the concurrent rise in risky drinking among both sexes, may increase the demand on the Norwegian healthcare resource allocation due to the elevated risk of alcohol-related conditions such as falls, injuries, adverse medication interactions, and chronic diseases [1]. Women may be of particular risk of the adverse events to alcohol consumption [2-4]. Similar trends in increased risky drinking among middle-aged and older adults have been observed in other countries [5-7] underscoring similar public health implications.

Moreover, read lines 562-566 on the same page:

Given that older adults with risky or harmful drinking patterns may still be active drivers, general practitioners could consider alcohol screening questionnaires and the use of PEth during the judgments of driver’s license renewals [8, 9]. They could also promote alternatives for safe transportation, especially in rural areas [8].

Lastly, read lines 570-772, on page 38:

Health authorities could integrate alcohol prevention efforts or campaigns into general practice, senior centers, voluntary organizations or through social media and print media targeting older adults.

[3] While the introduction discusses health-related harms of alcohol, it overlooks the well-documented connection between alcohol consumption and external harms such as risky driving on the road. Briefly Including this dimension of risk to other people would strengthen the framing of alcohol as a broader societal risk factor. Consider including these recent literatures, such as https://doi.org/10.1016/j.ijtst.2024.11.009, and https://doi.org/10.1080/23249935.2025.2516817 to strengthen the societal relevance of your work.

We appreciate your observation regarding the omission of the association between alcohol consumption and risky drinking in the Introduction. Thank you for recommending recently published literature, which has now been incorporated into the revised Introduction. Please read lines 99-102 on page 5:

It is a well-documented connection between alcohol consumption and external harms such as risky driving and crash injuries [8] with the potential to cause severe harm, not only to themselves but also to others, including pedestrians [8, 10, 11].

[4] The limitations of the paper, such as with the data and methodology and some other factors should be well acknowledged.

We appreciate your suggestions regarding the need to more thoroughly acknowledge the limitations in our data and methodology. In response, we have expanded the limitations section from line 598 on

page 39 to address potential biases that may affect the generalizability of our findings.

Thus, our findings may be influenced by non-response and selection biases, as the oldest individuals – particularly abstainers or heavy drinkers – may have declined participation due to poor health, resulting in a biased sample toward healthier individuals [12].

…

Compared to sales data [13, 14], self-reported use of alcohol is substantially lower, and most likely under-reported due to factors such as reporting bias and recall bias because of cognitive impairment, memory errors, social desirability bias, and/or stigma associated with alcohol consumption [15-17]. Gender role-related expectations and taboos surrounding substance use may contribute to sex-specific differences in the risk of misreporting alcohol consumption [18]. Under-reporting of alcohol consumption might have led to misclassification and underestimation of risky drinking and heavy episodic drinking in our study (lines 623-630 on page 40).

…

PEth is not susceptible to the inherent biases associated with self-reported alcohol consumption as PEth measures the actual ethanol intake during the last 2-4 weeks [19] (lines 633-634 on page 40).

Reviewer #2:

Strengths

1. The HUNT study is well explained, with survey waves, dates, and participation rates clearly documented.

2. The introduction is well written and provides clear context for the study.

3. The manuscript acknowledges both strengths (generalizability to Norway/Western health trends) and weaknesses (lack of large cities, fewer immigrants, lower education/income) of the Nord-Trøndelag population.

4. Use of questionnaires, registry data (Statistics Norway), and biobank samples is clearly stated.

5. Self-reports complemented with PEth biomarkers strengthens validity.

Minor issues

Indicate the type of model (logistic, multinomial, or linear) and link function should be stated.

Acknowledgement of potential biases (selection, multiple testing)

We appreciate these suggestions.

We have now included the required details regarding the type of the model in the Statistical analyses part of the manuscript, stating that the outcome variable was dichotomous, implying logistic regression model, and the logit link function which is a standard choice in the case of such an outcome variable. Please see lines 263-264 on page 15.

Please read lines 598-601 on page 39 regarding potential biases (also added as a response to reviewer 1):

Thus, our findings may be influenced by non-response and selection biases, as the oldest individuals – particularly abstainers or heavy drinkers – may have declined participation due to poor health, resulting in a biased sample toward healthier individuals [12].

…

Compared to sales data [13, 14], self-reported use of alcohol is substantially lower, and most likely under-reported due to factors such as reporting bias and recall bias because of cognitive impairment, memory errors, social desirability bias, and/or stigma associated with alcohol consumption [15-17]. Gender role-related expectations and taboos surrounding substance use may contribute to sex-specific differences in the risk of misreporting alcohol consumption[18]. Under-reporting of alcohol consumption might have led to misclassification and underestimation of risky drinking and heavy episodic drinking in our study (lines 623-630 on page 40).

…

PEth is not susceptible to the inherent biases associated with self-reported alcohol consumption as PEth measures the actual ethanol intake during the last 2-4 weeks [19] (lines 633-634 on page 40).

References

1. WHO. Global status report on alcohol and health and treatment of substance use disorders. Available from: https://www.who.int/publications/i/item/9789240096745o 2024.

2. Alfonso-Loeches S, Pascual M, Guerri C. Gender differences in alcohol-induced neurotoxicity and brain damage. Toxicology. 2013;311(1-2):27–34. Epub 20130314. doi: 10.1016/j.tox.2013.03.001. PubMed PMID: 23500890.

3. Bradley KA, Badrinath S, Bush K, Boyd-Wickizer J, Anawalt B. Medical risks for women who drink alcohol. Journal of general internal medicine. 1998;13(9):627–39. doi: 10.1046/j.1525-1497.1998.cr187.x. PubMed PMID: 9754520; PubMed Central PMCID: PMCPMC1497016.

4. Ceylan-Isik AF, McBride SM, Ren J. Sex difference in alcoholism: who is at a greater risk for development of alcoholic complication? Life Sci. 2010;87(5-6):133–8. Epub 20100616. doi: 10.1016/j.lfs.2010.06.002. PubMed PMID: 20598716; PubMed Central PMCID: PMCPMC2913110.

5. Ahlner F, Sigstrom R, Sterner TR, Fassberg MM, Kern S, Ostling S, et al. Increased alcohol consumption among Swedish 70-year-olds 1976 to 2016: Analysis of data from The Gothenburg H70 Birth Cohort Studies, Sweden. Alcoholism: Clinical and Experimental Research. 2018;42(12):2403–12. doi: http://dx.doi.org/10.1111/acer.13893. PubMed PMID: 2018-63313-013.

6. Bosque-Prous M, Espelt A, Sordo L, Guitart AM, Brugal MT, Bravo MJ. Job Loss, Unemployment and the Incidence of Hazardous Drinking during the Late 2000s Recession in Europe among Adults Aged 50-64 Years. PloS one. 2015;10(10):e0140017. Epub 20151007. doi: 10.1371/journal.pone.0140017. PubMed PMID: 26445239; PubMed Central PMCID: PMCPMC4596847.

7. Grucza RA, Sher KJ, Kerr WC, Krauss MJ, Lui CK, McDowell YE, et al. Trends in Adult Alcohol Use and Binge Drinking in the Early 21st‐Century United States: A Meta‐Analysis of 6 National Survey Series. Alcoholism: Clinical & Experimental Research. 2018;42(10):1939–50. doi: 10.1111/acer.13859. PubMed PMID: 132090333. Language: English. Entry Date: 20181004. Revision Date: 20181005. Publication Type: Article. Journal Subset: Biomedical.

8. Adeyemi O, Bukur M, Berry C, DiMaggio C, Grudzen CR, Konda S, et al. Substance use and pre-hospital crash injury severity among U.S. older adults: A five-year national cross-sectional study. PloS one. 2023;18(10):e0293138. Epub 20231025. doi: 10.1371/journal.pone.0293138. PubMed PMID: 37878571; PubMed Central PMCID: PMCPMC10599556.

9. Dyrkorn R, Skråstad RB, Aamo TO. [Bruk av fosfatidyletanol i førerkortsaker]. Use of phosphatidylethanol in driver’s license assessments. Only in Norwegian. Tidsskr Nor Laegeforen. 2019;139(3). Epub 20190211. doi: 10.4045/tidsskr.18.0973. PubMed PMID: 30754945.

10. Abdulrazaq MA, Fan WD. Temporal dynamics of pedestrian injury severity: A seasonally constrained random parameters approach . Available from: https://doi.org/10.1016/j.ijtst.2024.11.009 (11.09.2025). International Journal of Transportation Science and Technology. 2024.

11. Abdulrazaq MA, Fan WD. A priority based multi-level heterogeneity modelling framework for vulnerable road users. Available from: https://doi.org/10.1080/23249935.2025.2516817 (11.09.2025). Transportmetica A: Transport Science. 2025. doi: 10.1080/23249935.2025.2516817.

12. Goulden R. Moderate Alcohol Consumption Is Not Associated with Reduced All-cause Mortality. The American journal of medicine. 2016;129(2):180–6.e4. Epub 2015/11/03. doi: 10.1016/j.amjmed.2015.10.013. PubMed PMID: 26524703.

13. Høyer G, Nilssen O, Brenn T, Schirmer H. The Svalbard study 1988-89: a unique setting for validation of self-reported alcohol consumption. Addiction (Abingdon, England). 1995;90(4):539–44. doi: 10.1046/j.1360-0443.1995.9045397.x. PubMed PMID: 7773116.

14. Ramstedt M. How much alcohol do you buy? A comparison of self-reported alcohol purchases with actual sales. Addiction (Abingdon, England). 2010;105(4):649–54. Epub 20100209. doi: 10.1111/j.1360-0443.2009.02839.x. PubMed PMID: 20148793.

15. Kamsvaag B, Bergh S, Šaltytė Benth J, Selbaek G, Tevik K, Helvik AS. Alcohol consumption among older adults with symptoms of cognitive decline consulting specialist health care. Aging & mental health. 2022;26(9):1756–64. Epub 20210729. doi: 10.1080/13607863.2021.1950618. PubMed PMID: 34323134.

16. Moriconi PA, Nadeau L, Demers A. Drinking habits of older Canadians: a comparison of the 1994 and 2004 national surveys. Canadian journal on aging = La revue canadienne du vieillissement. 2012;31(4):379–93. Epub 2012/12/06. doi: 10.1017/s0714980812000347. PubMed PMID: 23211564.

17. Merrick EL, Horgan CM, Hodgkin D, Garnick DW, Houghton SF, Panas L, et al. Unhealthy drinking patterns in older adults: prevalence and associated characteristics. Journal of the American Geriatrics Society. 2008;56(2):214–23. Epub 2007/12/19. doi: 10.1111/j.1532-5415.2007.01539.x. PubMed PMID: 18086124.

18. Northcote J, Livingston M. Accuracy of self-reported drinking: observational verification of 'last occasion' drink estimates of young adults. Alcohol and alcoholism (Oxford, Oxfordshire). 2011;46(6):709–13. Epub 20110921. doi: 10.1093/alcalc/agr138. PubMed PMID: 21949190.

19. Skråstad RB, Spigset O, Aamo TO, Andreassen TN. Stability of Phosphatidylethanol 16:0/18:1 in Freshly Drawn, Authentic Samples from Healthy Volunteers. J Anal Toxicol. 2021;45(4):417–21. doi: 10.1093/jat/bkaa082. PubMed PMID: 32754728; PubMed Central PMCID: PMCPMC8040374.

---

## [Decision Letter · Decision Letter 1]

29 Sep 2025

Trends in alcohol consumption in middle-aged and older adults, assessed with self-report and the alcohol marker phosphatidylethanol - A longitudinal HUNT Study

PONE-D-25-31670R1

Dear Dr. Tevik,

We’re pleased to inform you that your manuscript has been judged scientifically suitable for publication and will be formally accepted for publication once it meets all outstanding technical requirements.

Kind regards,

Y-h. Taguchi, Dr. Sci.

Academic Editor

PLOS ONE

Additional Editor Comments (optional):

The paper was accepted.

Reviewers' comments:

Reviewer's Responses to Questions

**Comments to the Author**

1. If the authors have adequately addressed your comments raised in a previous round of review and you feel that this manuscript is now acceptable for publication, you may indicate that here to bypass the “Comments to the Author” section, enter your conflict of interest statement in the “Confidential to Editor” section, and submit your "Accept" recommendation.

Reviewer #1: All comments have been addressed

2. Is the manuscript technically sound, and do the data support the conclusions?

Reviewer #1: Yes

3. Has the statistical analysis been performed appropriately and rigorously? 

Reviewer #1: Yes

4. Have the authors made all data underlying the findings in their manuscript fully available?

Reviewer #1: Yes

5. Is the manuscript presented in an intelligible fashion and written in standard English?

Reviewer #1: Yes

6. Review Comments to the Author

Reviewer #1: (No Response)

7. PLOS authors have the option to publish the peer review history of their article (what does this mean? ). If published, this will include your full peer review and any attached files.

**Do you want your identity to be public for this peer review?** For information about this choice, including consent withdrawal, please see our Privacy Policy .

Reviewer #1: No

---

## [Editor Report · Acceptance letter]

PONE-D-25-31670R1

PLOS ONE

Dear Dr. Tevik,

I'm pleased to inform you that your manuscript has been deemed suitable for publication in PLOS ONE. Congratulations! Your manuscript is now being handed over to our production team.

Kind regards,

on behalf of

Professor Y-h. Taguchi

Academic Editor

PLOS ONE